# Latent Diffusion for Language Generation

**Justin Lovelace**[*]    **Varsha Kishore**    **Chao Wan**    **Eliot Shekhtman**    **Kilian Q. Weinberger**
Cornell University, Ithaca, NY

## Abstract

Diffusion models have achieved great success in modeling continuous data modalities such as images, audio, and video, but have seen limited use in discrete domains such as language. Recent attempts to adapt diffusion to language have presented diffusion as an alternative to existing pretrained language models. We view diffusion and existing language models as complementary. We demonstrate that encoder-decoder language models can be utilized to efficiently learn high-quality language autoencoders. We then demonstrate that continuous diffusion models can be learned in the latent space of the language autoencoder, enabling us to sample continuous latent representations that can be decoded into natural language with the pretrained decoder. We validate the effectiveness of our approach for unconditional, class-conditional, and sequence-to-sequence language generation. We demonstrate across multiple diverse data sets that our latent language diffusion models are significantly more effective than previous diffusion language models. Our code is available at `https://github.com/justinlovelace/latent-diffusion-for-language`.

## 1 Introduction

Although originally introduced by Sohl-Dickstein et al. [61] in 2015, diffusion models did not see widespread use until Ho et al. [22] demonstrated their viability for high-quality image generation in 2020. Since then, research has driven rapid improvements and they have recently surpassed generative adversarial networks on image generation benchmarks [12] and autoregressive models on density estimation benchmarks [30], outclassing generative modeling paradigms that have dominated those areas for the better part of a decade. Diffusion models are now, arguably, the most widely used class of generative models for continuous data modalities such as images, audio, and video [54, 32, 23].

The widespread success of diffusion models across a variety of domains and applications makes them appealing for language generation. However, they have seen less use in discrete domains, where the gradual transition of discrete states to Gaussian noise (and vice versa) is not as natural as in continuous domains. Prior work proposes to learn continuous diffusion models in the space of learnable word embeddings and decodes the continuous generations with a rounding step [36]. However, combining representation learning with the diffusion objective requires careful regularization to avoid collapse.

One breakthrough in image generation was the introduction of latent diffusion [51], where diffusion models are trained to produce samples from the latent distribution of a pretrained autoencoder. This offloads the task of generating high-frequency details to the autoencoder and enables the diffusion process to focus on the high-level semantics of images. In this paper, we explore the viability of latent diffusion for text generation. We claim that this approach is particularly well-suited for discrete modalities because it offloads the challenge of modeling a discrete distribution to the autoencoder and simplifies the diffusion process by restricting it to the continuous, latent feature space.

We introduce Latent Diffusion for Language Generation (LD4LG), a method that leverages the latent space of a pretrained encoder-decoder network (e.g. BART [35], T5 [50]) to learn a high-quality

---

[*]Correspondence to <jl3353@cornell.edu>.

37th Conference on Neural Information Processing Systems (NeurIPS 2023).

diffusion model for text. The latent representations from such models are high-dimensional and input-length dependent — complicating the use of diffusion models [51, 66]. To address both issues, we learn an additional compression module that maps the high-dimensional encoder representations to a lower-dimensional fixed-length representation. We also learn a corresponding reconstruction network to map these fixed-length features back to high dimensional features that guide the language decoder (via cross-attention) to reconstruct the original language.

The low-dimensional representation is ideally suited for diffusion. For language generation, we use a diffusion model to generate a low-dimensional (fixed-length) latent, which is mapped into a higher dimensional space with the reconstruction network. This high dimensional representation then guides the pre-trained decoder to generate natural language. Our approach naturally combines the continuous, fixed-length diffusion process with discrete, variable length text generation.

We demonstrate that LD4LG is effective for unconditional, class-conditional, and sequence-to-sequence language generation across a variety of datasets. Our approach significantly outperforms recent diffusion language models while using fewer sampling steps. For instance, we achieve a MAUVE score [47] of .716 for the ROCStories dataset with 250 sampling steps while Diffusion-LM [36] achieves a MAUVE score of .043 using 2000 sampling timesteps. For the challenging XSum summarization benchmark, we achieve a ROUGE-L of 31.9 with 250 timesteps while the recently proposed DiffuSeq [16] achieves a ROUGE-L of 14.1 with 2000 timesteps. We also find that the diffusion models offer some benefits over a strong autoregressive baseline. In particular, we observe that our latent language diffusion is less susceptible to memorization and more effective for class-conditional generation.

## 2 Background

Diffusion models [61, 22, 63] are a class of latent variable models that learn to iteratively transform random Gaussian noise, which can be sampled analytically, to a sample from an unknown data distribution specified by a collection of samples. This mapping is defined through a forward diffusion process that iteratively adds Gaussian noise to samples, and a generative process that iteratively "denoises" samples from the Gaussian distribution to obtain samples from the data distribution. We provide a formal description of diffusion models in the appendix.

The diffusion model consists of a denoising network $\hat{\mathbf{x}}_\theta$ trained with a regression objective

$$\mathcal{L}(\theta) = \mathbb{E}_{t,\mathbf{x},\epsilon}[\lambda_t \|\hat{\mathbf{x}}_\theta(\sqrt{\alpha_t}\mathbf{x} + \sqrt{1 - \alpha_t}\epsilon, t) - \mathbf{x}\|_2^2]$$

where $\mathbf{x}$ is the training data, $t \sim \mathcal{U}(\mathbf{0}, \mathbf{1})$ is the timestep, $\epsilon \sim \mathcal{N}(\mathbf{0}, \mathbf{1})$ is Gaussian noise, $\alpha_t$ defines the noise schedule, and $\lambda_t$ is a time-dependent weighting term. The denoising network is therefore trained to denoise a noisy latent, $\mathbf{z}_t = \sqrt{\alpha_t}\mathbf{x} + \sqrt{1 - \alpha_t}\epsilon$, to the clean data, $\mathbf{x}$, with a regression objective that emphasizes certain times $t$. Sampling algorithms start from pure Gaussian noise, $\mathbf{z}_1 \sim \mathcal{N}(\mathbf{0}, \mathbf{1})$, and utilize the denoising network to iteratively generate latents $\mathbf{z}_{t_1}, \mathbf{z}_{t_2}, ..., \mathbf{z}_{t_T}$ where $1 = t_1 > t_2 > ... > t_T = 0$, with decreasing levels of noise until $\mathbf{z}_0$ is drawn approximately from the data distribution.

## 3 Latent Diffusion For Language

Figure 1 presents an overview of Latent Diffusion for Language Generation. Our method consists of two main parts. We augment a pretrained encoder-decoder language model with two learnable networks to develop a high-quality language autoencoder with a compact latent space. We then introduce continuous diffusion models that learn to generate samples from the latent distribution of our language autoencoders. These continuous samples can, by design, be decoded into natural language.

### 3.1 Language Autoencoder

We base our architecture on pretrained encoder-decoder language models (depicted in blue), such as BART [35] and T5 [50] (we present results with both). By default, we freeze the pre-trained models and learn only the autoencoding modules to accelerate training. The *Language Encoder*, $E(\cdot)$, maps variable-length language, represented as a sequence of tokens, $\mathbf{w} \in \mathbb{N}^L$, to a latent representation of the same length, $E(\mathbf{w}) \in \mathbb{R}^{L \times d_{\text{LM}}}$.

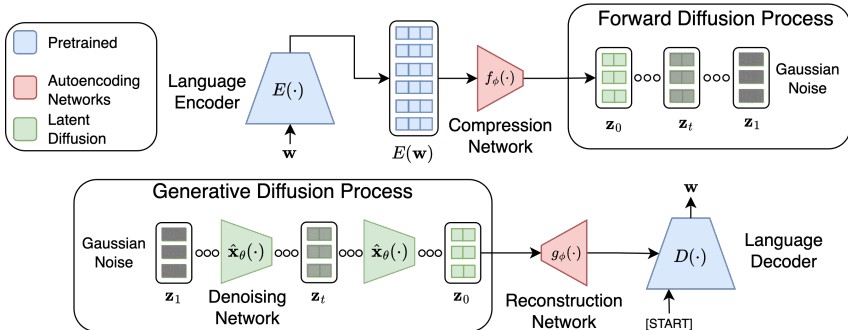

Figure 1: Overview of our proposed latent language diffusion framework.

**Compression Network.** The learnable *Compression Network* maps the encoder features to a compact latent space that is well-suited for diffusion. We adopt the Perceiver Resampler [2] architecture, originally developed to compress image features for a vision-language model, which is depicted in Figure 2. The Perceiver Resampler, like the transformer, consists of a stack of alternating multi-head attention (MHA) blocks and feedforward (FF) layers. We refer the reader to Vaswani et al. [69] for a detailed description of these components. We learn $\ell$ latent queries $Z \in \mathbb{R}^{\ell \times d_{\text{LM}}}$ that iteratively cross-attend to the language encoder features $E(\mathbf{w}) \in \mathbb{R}^{L \times d_{\text{LM}}}$ to extract information. We follow Alayrac et al. [2] and allow the latent queries to simultaneously attend to themselves and the frozen encoder representations. We can write the attention layer as

$$Z = Z + \text{MHA}(\text{q} = Z, \text{kv} = [Z; E(\mathbf{w})])$$

where MHA$(\cdot)$ is the multi-head attention operation with queries, q, and keys/values, kv. This design compresses the encoder representations to the fixed sequence length, $\ell$, of the latents. After each multi-head attention layer, a feedforward layer is applied to the latent query representations.

After the Compression Network maps the input to a fixed sequence length, we reduce the dimensionality of the output to dimension $d_{\text{ae}}$ with a learnable linear projection. The compression network therefore maps the variable length output of the frozen encoder to a compact latent space

$$\mathbf{x} = f_\phi(E(\mathbf{w})) \in \mathbb{R}^{\ell \times d_{\text{ae}}}$$

of fixed length $\ell < L$ and dimensionality $d_{\text{ae}} < d_{\text{LM}}$ where we will learn our diffusion model.

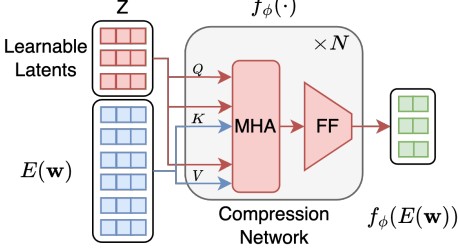

Figure 2: Architecture of our Compression Network.

To ensure that the latent space is appropriately scaled for diffusion, we can optionally constrain the norm of the latent space. Since $\mathbb{E}_{\epsilon \sim \mathcal{N}(\mathbf{0},\mathbf{I})}[\|\epsilon\|_2^2] = d_{\text{ae}}$ [1], we can normalize the latent vectors along the feature dimension so that $\|\mathbf{x}_i\|_2^2 = d_{\text{ae}}$ similar to prior work on text diffusion [13].

**Reconstruction Network.** The *Reconstruction Network* maps the compressed latent space to the feature space expected by the *Language Decoder*. To achieve this, we project $\mathbf{x} = f_\phi(E(\mathbf{w})) \in \mathbb{R}^{\ell \times d_{\text{ae}}}$ back up to dimension $d_{\text{LM}}$, add learnable absolute position embeddings, and pass it through a standard transformer model to obtain features $g_\phi(\mathbf{x}) \in \mathbb{R}^{\ell \times d_{\text{LM}}}$.

The *Language Decoder*, $D(\cdot)$, cross-attends to to these features and generates text autoregressively. We train the compression and reconstruction networks to produce features that guide the decoder to reconstruct the input text

$$\mathbf{w} \approx D(g_\phi(\mathbf{x})) = D(g_\phi(f_\phi(E(\mathbf{w}))))$$

with the cross-entropy loss. This gives us a continuous, semantic latent space that can be decoded to natural language.

**Implementation Details.** We utilize BART-base and FLAN-T5-base [10] as the encoder-decoder language models throughout this work and learn language autoencoders for each dataset. During autoencoder training, we freeze the pre-trained language models and only learn the autoencoding modules. The autoencoder training could likely be amortized across datasets by training a general-purpose language autoencoder on a large corpus of text, but we leave such explorations to future work. We train the autoencoder to reconstruct the input language with the cross-entropy loss. For the diffusion latent space, we set $\ell = 32, d_{\mathrm{ae}} = 64$ and utilize 3 layers in both autoencoding modules across all monolingual datasets.

For our machine translation experiments, we utilize MT5-base [72] to develop our autoencoder. We found it beneficial to jointly fine-tune the language model and the autoencoding modules, likely because the dataset is an order of magnitude larger than our other datasets and therefore benefits from the additional capacity. We use the same latent dimensionality, but only use a single layer for the autoencoding modules. We report full hyperparameter settings in the appendix. We constrain the norm of the latent space across models and datasets except when using FLAN-T5 because it led to a minor degradation in autoencoding performance and downstream generation quality.

## 3.2 Latent Language Diffusion

Figure 1 outlines our latent language diffusion framework. Given some dataset of natural language, $\mathcal{D}$, we can now sample continuous data as $\mathbf{x} = f_\phi(E(\mathbf{w})) \in \mathbb{R}^{\ell \times d_{\mathrm{ae}}}$ where $\mathbf{w} \sim \mathcal{D}$. We then train a continuous denoising network, $\hat{\mathbf{x}}_\theta()$, to recover $\mathbf{x}$ with the standard regression objective

$$\mathcal{L}(\theta) = \mathbb{E}_{t,\mathbf{x},\epsilon}[\lambda_t \|\hat{\mathbf{x}}_\theta(\sqrt{\alpha_t}\mathbf{x} + \sqrt{1-\alpha_t}\epsilon, t) - \mathbf{x}\|_2^2]$$

with some time-dependent weighting $\lambda_t$. In practice, the denoising network is often parameterized as an $\epsilon$-prediction network [22] or a $\mathbf{v}$-prediction network [57] where the velocity, $\mathbf{v}$, is defined as $\mathbf{v} = \sqrt{\alpha_t}\epsilon - \sqrt{1-\alpha_t}\mathbf{x}$. These parameterizations can be interpreted as different weighting functions, $\lambda_t$, for the regression objective above (see Salimans and Ho [57]. We adopt the $\mathbf{v}$-parameterization in this work because it has been shown to be effective for latent image diffusion [51].

For generation, we sample a latent variable, $\mathbf{z}_1 \in \mathbb{R}^{\ell \times d_{\mathrm{ae}}} \sim \mathcal{N}(\mathbf{0}, \mathbf{I})$, that is iteratively denoised to produce a sample, $\mathbf{x} = \mathbf{z}_0$, from the distribution of the language autoencoder's latent space. We then generate natural language with the pretrained reconstruction network and language decoder $\mathbf{w} = D(g_\phi(\mathbf{x}))$. We train our diffusion models with the cosine noise schedule $\alpha_t = \cos(0.5\pi t)^2$ [45, 57, 55] by default. For our machine translation experiments, we employ a scaled cosine noise schedule (see subsection E.2 in the appendix for full details) [7, 27]. For generation, we use the DDPM sampler with 250 sampling timesteps. For text generation with the pretrained decoder, we utilize beam search with 4 beams. We train all of our diffusion models with a single Nvidia A6000 GPU except for the machine translation models which are trained with 4 Nvidia A6000 GPUs.

**Denoising Network Architecture.** Our denoising network, $\hat{\mathbf{x}}_\theta(\mathbf{z}_t, t)$, is a pre-LayerNorm transformer [69, 70] with 12 layers and a dimension of 768. We utilize learnable absolute positional encodings and GeGLU activations [59]. Bao et al. [4] adapted transformers to image diffusion and found that dense connections [28] between early and late layers are beneficial due to the dense nature of the denoising objective. We adopt this modification to improve the suitability of the transformer for diffusion. The autoencoder latent is projected to the transformer dimension, processed by the transformer, and then projected back to dimensionality of the autoencoder latent to obtain the final prediction. Following prior work [6, 56, 7], we utilize $\alpha$-conditioning to condition the model on the level of noise. We map $\alpha_t$ to a sinusoidal positional embedding [69] and pass it through an MLP with a single hidden layer to obtain a time embedding. We add this time embedding to the input sequence and apply adaptive layer normalization [46] conditioned on the time embedding to the output of every feedfoward layer.

**Self-Conditioning** We utilize the self-conditioning technique introduced by Chen et al. [8] which has been shown to improve the quality of diffusion models [8, 67]. The denoising network is typically conditioned on the latent variable and the current timestep as $\tilde{\mathbf{x}}_t = \hat{\mathbf{x}}_\theta(\mathbf{z}_t, t)$. Self-conditioning proposes to condition the network on its estimate of the data from the previous timestep, $s > t$, to improve the prediction at the current timestep $\tilde{\mathbf{x}}_t = \hat{\mathbf{x}}_\theta(\mathbf{z}_t, t, \tilde{\mathbf{x}}_s)$. During inference, the sampling procedure is inherently iterative and at time $t$, we have already computed the output of the denoising network for the previous step. Therefore, it does not require any additional applications of the

network. We must, however, modify the training procedure so that the denoising network learns to utilize the estimate of the data, and we must define the inference behavior for the first timestep.

For each training step, we sample some time $t \sim \mathcal{U}([0,1])$ as before. With probability $p$, we do not provide any estimate of the data for self-conditioning, denoted $\tilde{\mathbf{x}}_{t,\emptyset} = \hat{\mathbf{x}}_\theta(\mathbf{z}_t, t, \emptyset)$. With probability $1 - p$, however, we mimic the inference behavior by first computing $\tilde{\mathbf{x}}_{t,\emptyset} = \hat{\mathbf{x}}_\theta(\mathbf{z}_t, t, \emptyset)$ and then computing an additional estimate $\tilde{\mathbf{x}}_t = \hat{\mathbf{x}}_\theta(\mathbf{z}_t, t, \text{sg}(\tilde{\mathbf{x}}_{t,\emptyset}))$ where $\text{sg}()$ is the stop-gradient operation. This second estimate is then used to compute the loss. We follow Chen et al. [8] and set $p = 0.5$.

This training procedure also maintains the capacity for inference without self-conditioning which is utilized to generate the first estimate during sampling. We condition on the previous estimate by concatenating it with the noisy latent along the feature dimension. When the previous estimate is not provided, we concatenate a learnable embedding with the noisy latent.

**Class-Conditional Diffusion.** For class-conditional diffusion, we have some dataset where each natural language utterance is associated with one of $C$ class labels representing, for example, the topic of the text. We condition the denoising network on the class label, $y$, during training, $\tilde{\mathbf{x}}_t = \hat{\mathbf{x}}_\theta(\mathbf{z}_t, t, y)$. We replace the ground truth class label, $y_i$, with a null label, $y_\emptyset$, with probability $p = 0.1$ to maintain the capacity for unconditional generation. At inference time, we can choose some class $y$ to guide the sampling process to generate text from the specified class. We condition on class labels by introducing learnable embeddings for all labels, including the null label, and add it to the time embedding.

**Sequence-to-Sequence Diffusion.** Given some seq2seq dataset consisting of source-target language pairs $(\mathbf{w}_{\text{src}}, \mathbf{w}_{\text{trg}}) \sim \mathcal{D}$, we condition our denoising network on the source sequence and generate the target latent $\mathbf{x}_{\text{trg}} = f_\phi(E(\mathbf{w}_{\text{trg}}))$. For news summarization, for instance, we generate a latent representation of the summary by conditioning the network on the article text. To achieve this, we introduce a cross-attention layer after every self-attention layer in the denoising network that attends to features from a frozen language encoder.

In general, we can incorporate any language encoder, $E_{\text{src}}(\cdot)$, to extract features from the source text. By default, we use the same pretrained encoder used for our language autoencoder. For our machine translation experiments, we condition our latent diffusion models on representations from a frozen MT5-XL encoder, which we found to be more effective than MT5-base representations. Therefore, given a sample from our seq2seq dataset, $(\mathbf{w}_{\text{src}}, \mathbf{w}_{\text{trg}}) \sim \mathcal{D}$, we can compute $\mathbf{x}_{\text{trg}} = f_\phi(E(\mathbf{w}_{\text{trg}}))$ and use a modified seq2seq diffusion objective

$$\mathcal{L}(\theta) = \mathbb{E}_{t,(\mathbf{w}_{\text{src}},\mathbf{w}_{\text{trg}}),\epsilon}[\lambda_t \|\hat{\mathbf{x}}_\theta(\sqrt{\alpha_t}\mathbf{x}_{\text{trg}} + \sqrt{1 - \alpha_t}\epsilon, t, E_{\text{src}}(\mathbf{w}_{\text{src}})) - \mathbf{x}_{\text{trg}}\|_2^2].$$

We also utilize classifier-free guidance [21] to improve sample quality. We jointly learn an unconditional network, $\hat{\mathbf{x}}_\theta(\mathbf{z}_t, t)$, and a conditional network, $\hat{\mathbf{x}}_\theta(\mathbf{z}_t, t, E(\mathbf{w}_{\text{src}}))$, by dropping the conditioning information with probability $p = 0.1$ during training. When we drop the conditioning information, we cross-attend to a learnable embedding instead of the embedded source text. During sampling, we use guidance weight $w$ and compute the prediction as

$$\tilde{\mathbf{x}}_t = w\hat{\mathbf{x}}_\theta(\mathbf{z}_t, t, E(\mathbf{w}_{\text{src}})) + (1 - w)\hat{\mathbf{x}}_\theta(\mathbf{z}_t, t).$$

Setting $w = 1.0$ corresponds to the conditional diffusion model while setting $w > 1.0$ strengthens the influence of the conditioning information. We use $w = 2.0$ for the seq2seq tasks and ablate this choice in section 5.

We can also generate multiple outputs $\mathcal{S}$ for each input by sampling different latents $\mathbf{z}_1 \sim \mathcal{N}(\mathbf{0}, \mathbf{I})$. We then select the most promising candidate with Minimum Bayes Risk (MBR) Decoding [15, 34]. In MBR decoding, we define a loss function $\mathcal{L}$, such as the negative Rouge, and use it to select a candidate $\mathbf{w}_{\text{MBR}} = \text{argmin}_{\mathbf{w}\in\mathcal{S}}\frac{1}{|\mathcal{S}|}\sum_{\mathbf{w}'\in\mathcal{S}}\mathcal{L}(\mathbf{w}, \mathbf{w}')$. In our experiments, we use $|\mathcal{S}| = 5$ and denote the results from using MBR decoding as MBR-5. We also report results using the ground truth to select the best candidate $\mathbf{w}_{\text{oracle}} = \text{argmin}_{\mathbf{w}\in\mathcal{S}}\mathcal{L}(\mathbf{w}, \mathbf{w}_{\text{trg}})$ to provide an upper bound on the performance of our method given optimal sample selection. Because this requires knowledge of the ground-truth target text, we refer to this as Oracle sampling.

## 4   Datasets

We evaluate LD4LG on a variety of natural language datasets. **ROCStories** [42] is a corpus of 98k five-sentence commonsense stories, that capture casual and temporal relations. The **AG News Topic**

**Classification** [60] dataset consists of news articles across four topics: World, Sports, Business, Sci/Tech with article titles and descriptions from 120k training instances. We focus on generating the article descriptions in this work. The **XSum** [44] dataset consists of BBC articles from 2010 to 2017 covering a wide range of topics (e.g., News, Politics, Sports, etc.). The training split has 204k instances and each example contains a document and a summary. The **QQP** [9] dataset consists of 400k question pairs, where each example is two similar questions and a binary value indicating whether the two questions have the same meaning. The **WMT 2014 English-German** [5] dataset is a widely used machine translation dataset consisting of roughly 4.5 million sentence pairs. We present detailed dataset statistics in the appendix.

### 4.1 Evaluation Metrics.

We use **MAUVE Score** [47] and **Perplexity** (Ppl) to evaluate the quality of our generated text. MAUVE Score is a metric for open-ended text generation that compares the distribution of generated text with that of reference text using divergence frontiers. We follow Pillutla et al. [47] and use the GPT-2-Large model [49] to embed the text. Perplexity measures how likely the generated samples are according to an autoregressive language model; we use GPT-2-Large to compute perplexity.

We also want to quantify the **Diversity** (Div) of generations. We define diversity as $\mathbf{Div} = \prod_{n=2}^{4} \frac{|\text{unique n-grams}(\{\mathbf{w}_i\})|}{|\text{total n-grams}(\{\mathbf{w}_i\})|}$ where $\{\mathbf{w}_i\}$ is a set of generated samples [68]. The metrics discussed so far can be optimized by generating samples from the training set. We measure the proportion of generated 4-grams that are found in the training set to quantify the degree of **Memorization** (Mem).

To evaluate the performance for monolingual seq2seq language generation tasks, we utilize **Rouge** [37] and **BERTScore** [75]. Rouge-1/2 measures the number of unigrams/bigrams in the reference that appear in the generated text and Rouge-L measures the longest common sequence between the texts. BERTScore uses contextual embeddings from a pretrained language model to measure the similarity between texts. We follow prior work and use the `microsoft/deberta-xlarge-mnli` model [18] to extract contextual embeddings. For our machine translation experiments, we report **SacreBLEU** scores [5] to ensure fair comparison with prior work.

For our unconditional and class-conditional language generation experiments, we sample 1000 instances from the diffusion model. For the MAUVE reference text, we sample 1000 instances from the test set. We repeat this 5 times and report the mean and standard deviation as mean$_{\text{stdev}}$. We also compute reference values for our metrics with natural samples from the test set. The reference MAUVE, for instance, is computed between 1000 train and 1000 test samples. Qualitative samples from our models are in the supplemental materials.

## 5 Experiments

### 5.1 Language Autoencoder

We evaluate the effectiveness of our proposed language autoencoder using heldout examples from our datasets. As a point of comparison, we also evaluate the default behavior of the language models that we use to develop the language autoencoders. A consequence of BART's particular denoising objective is that the pretrained model already generates a copy of the input language, although this is not true of other models such as T5 or FLAN-T5.

We present results for our two most complex datasets, ROCStories and AG News, in Table 1 and present the results for XSum, QQP, and WMT14-En-De, which show similar trends, in the appendix. We observe that our BART-base autoencoder is able to compress the feature space by a factor of $24\times$ while improving the fidelity of the reconstructions. Our autoencoding modules are also effective at converting the pretrained FLAN-T5 into a language autoencoder, even though that is different from the model's default behavior. Across both models and all datasets, our language autoencoders are able to achieve near-perfect reconstruction with a low-dimensional latent space.

### 5.2 Unconditonal Language Generation

**Baselines.**  We evaluate our approach's capacity for unconditional language generation with the ROCStories and AG News datasets. We compare against the recently proposed Diffusion-LM model

Table 1: Effectiveness of Language Autoencoder

| Method | Latent Dimensions | Hidden Units | RocStories | | AG News | |
|---|---|---|---|---|---|---|
| | | | Rouge-1/2/L | BLEU | Rouge-1/2/L | BLEU |
| BART-Base | $L \times 768$ | $\leq 49{,}152$ | 98.9/98.2/98.8 | 97.5 | 99.6/99.4/99.6 | 98.6 |
| BART-Base Autoencoder | $32 \times 64$ | 2048 | 99.2/98.5/99.2 | 97.6 | 99.7/99.4/99.7 | 98.8 |
| FLAN-T5-Base | $L \times 768$ | $\leq 49{,}152$ | 21.5/11.8/19.4 | 0.7 | 63.6/53.0/59.6 | 42.3 |
| FLAN-T5-Base Autoencoder | $32 \times 64$ | 2048 | 98.4/96.9/98.4 | 95.8 | 99.1/98.3/99.1 | 96.8 |

[36]. We also fine-tune the pretrained GPT-2-Medium model, which is roughly $1.6\times$ larger than our denoising network, as a strong autoregressive baseline [49]. For sampling from GPT-2, we prompt it with a BOS token and utilize nucleus sampling ($p = 0.95$) [24]. We explore different sampling configurations in the appendix and find that they lead to similar conclusions.

**Results.** We present this comparison in Table 2. We observe that our approach is significantly more effective than Diffusion-LM at modeling language distributions, as demonstrated by the higher MAUVE scores, while requiring fewer sampling steps. Diffusion-LM is unable to model diverse language distributions and exhibits poor diversity. Utilizing high quality latent spaces from pretrained language models improves the effectiveness of our diffusion model. We observe that both language models are highly effective for the AG News dataset, but using BART-base leads to a stronger MAUVE score for the ROCStories dataset. Across both datasets, FLAN-T5-base produces more diverse generations and exhibits less memorization.

While GPT-2 generally achieves strong language generation metrics, it is more susceptible to memorization than LD4LG. For the AG News dataset, GPT-2 exhibits significant memorization and a lower MAUVE score. We do find that GPT-2 samples have lower perplexity. However, measuring perplexity with a pretrained GPT-2 model likely biases the metric towards the fine-tuned GPT-2 model. Moreover, MAUVE scores have a stronger correlation with human judgments of quality [47].

Table 2: Unconditional Language Generation Evaluation. The fine-tuned language model is presented in gray.

| | Timesteps | ROCStories | | | | AG News | | | |
|---|---|---|---|---|---|---|---|---|---|
| | | MAUVE ↑ | Ppl ↓ | Div ↑ | Mem ↓ | MAUVE ↑ | Ppl ↓ | Div ↑ | Mem ↓ |
| Reference | - | $.951_{.007}$ | $21.1_{.3}$ | $.414_{.003}$ | $.362_{.003}$ | $.951_{.014}$ | $43.6_{1.2}$ | $.658_{.002}$ | $.385_{.005}$ |
| Diffusion-LM [36] | 2000 | $.043_{.006}$ | $47.3_{.6}$ | $.128_{.002}$ | $.434_{.002}$ | $.012_{.001}$ | $67.1_{1.2}$ | $.043_{.002}$ | $.086_{.006}$ |
| LD4LG (BART-Base) | 250 | $.716_{.019}$ | $30.6_{.5}$ | $.331_{.005}$ | $.441_{.004}$ | $.866_{.016}$ | $100.6_{2.9}$ | $.540_{.006}$ | $.293_{.001}$ |
| LD4LG (FLAN-T5-base) | 250 | $.481_{.007}$ | $37.5_{.4}$ | $.389_{.002}$ | $.387_{.002}$ | $.859_{.020}$ | $122.0_{3.9}$ | $.624_{.008}$ | $.221_{.003}$ |
| GPT-2-Medium | - | $.788_{.025}$ | $20.0_{.2}$ | $.372_{.002}$ | $.688_{.006}$ | $.820_{.012}$ | $37.3_{1.1}$ | $.532_{.017}$ | $.829_{.005}$ |

**Benefits of Compression.** Because the pretrained BART model already copies the input text, we can ablate the impact of learning a compact latent space by learning a diffusion model directly in the encoder feature space. One complication of this setting is that the sequence length of the BART features vary. During training, the sequence length is simply determined by the sample. During generation, however, we must specify the length. To determine the sequence length for generation, we opt to sample a length from the empirical distribution of lengths in the training set. We refer to this baseline as BART-Diffusion and outline full implementation details in the appendix.

We compare BART-Diffusion with our proposed approach in Table 3. We quantify the speedup by measuring how long it takes each approach to match the peak validation MAUVE of BART-Diffusion. We observe that learning a compact latent space is beneficial both in terms of absolute performance and wall-clock time, reaching the peak MAUVE of BART-diffusion in a quarter of the time. Compressing the latent space along the sequence dimension significantly reduces the overhead per iteration due to the quadratic cost of self-attention, and we also observe faster convergence.

**Self-conditioning.** We ablate the impact of self-conditioning in Table 4. We find that it significantly improves the MAUVE score and the perplexity of the generated text, but sacrifices some diversity.

Table 3: Benefits of Compression (ROCStories)

| | Hidden Units | Relative Speedup | MAUVE ↑ | Ppl ↓ | Div ↑ | Mem ↓ |
|---|---|---|---|---|---|---|
| BART-Diffusion | $\leq$ 49,152 | 1.0× | $.605_{024}$ | $46.8_{7}$ | $.424_{004}$ | $.304_{003}$ |
| LD4LG (BART-base) | 2048 | 3.86× | $.716_{019}$ | $30.6_{5}$ | $.331_{005}$ | $.441_{004}$ |

Table 4: Impact of Self-Conditioning (ROCStories)

| | MAUVE ↑ | Ppl ↓ | Div ↑ | Mem ↓ |
|---|---|---|---|---|
| LD4LG (BART-base) | $.716_{019}$ | $30.6_{5}$ | $.331_{005}$ | $.441_{004}$ |
| - Self-cond. | $.480_{018}$ | $79.3_{1.0}$ | $.427_{004}$ | $.299_{003}$ |

Table 5: Metrics for class-conditional generation.

| | | LD4LG (BART-base) | | | | | LD4LG (FLAN-T5-base) | | | | |
|---|---|---|---|---|---|---|---|---|---|---|---|
| | | MAUVE ↑ | | | | Mem ↓ | MAUVE ↑ | | | | Mem ↓ |
| | Conditioning | World | Sports | Business | Sci/Tech | | World | Sports | Business | Sci/Tech | |
| Diffusion | World | $.842_{017}$ | $.015_{002}$ | $.026_{002}$ | $.020_{002}$ | $.296_{002}$ | $.809_{024}$ | $.013_{001}$ | $.025_{002}$ | $.022_{002}$ | $.233_{005}$ |
| | Sports | $.013_{001}$ | $.845_{024}$ | $.011_{001}$ | $.010_{000}$ | $.305_{003}$ | $.011_{001}$ | $.836_{020}$ | $.009_{000}$ | $.009_{000}$ | $.249_{004}$ |
| | Business | $.024_{002}$ | $.011_{001}$ | $.752_{030}$ | $.068_{005}$ | $.363_{009}$ | $.025_{003}$ | $.011_{001}$ | $.765_{016}$ | $.076_{008}$ | $.244_{004}$ |
| | Sci/Tech | $.023_{002}$ | $.012_{001}$ | $.082_{008}$ | $.813_{028}$ | $.225_{004}$ | $.024_{001}$ | $.011_{001}$ | $.082_{010}$ | $.843_{033}$ | $.169_{004}$ |

| | | Conditional GPT-2 | | | | | Reference | | | | |
|---|---|---|---|---|---|---|---|---|---|---|---|
| | | MAUVE ↑ | | | | Mem ↓ | MAUVE ↑ | | | | Mem ↓ |
| | Conditioning | World | Sports | Business | Sci/Tech | | World | Sports | Business | Sci/Tech | |
| Comparisons | World | $.805_{022}$ | $.012_{000}$ | $.025_{002}$ | $.021_{002}$ | $.402_{002}$ | $.963_{009}$ | $.018_{001}$ | $.034_{002}$ | $.032_{003}$ | $.388_{007}$ |
| | Sports | $.017_{001}$ | $.840_{019}$ | $.012_{001}$ | $.013_{001}$ | $.369_{004}$ | $.018_{001}$ | $.958_{007}$ | $.014_{001}$ | $.014_{002}$ | $.346_{002}$ |
| | Business | $.037_{003}$ | $.012_{001}$ | $.629_{029}$ | $.069_{007}$ | $.479_{007}$ | $.040_{005}$ | $.014_{001}$ | $.968_{009}$ | $.125_{009}$ | $.441_{003}$ |
| | Sci/Tech | $.033_{002}$ | $.013_{001}$ | $.102_{015}$ | $.697_{027}$ | $.434_{004}$ | $.036_{003}$ | $.016_{001}$ | $.133_{013}$ | $.955_{011}$ | $.366_{003}$ |

## 5.3 Class-Conditonal Language Generation

**Baselines.** Conditional training with control tokens is one of the most widely used methods for controlling autoregressive models [14, 29, 40, 33]. We prepend the class label to each sample as a control token and fine-tune GPT-2-medium for class-conditional generation. Because memorizing the training instances associated with each class is a trivial solution, we terminate training when the model's memorization exceeds the reference values.

**Results.** We evaluate the effectiveness of class-conditioning with the AG News topic classification dataset. We sample instances for each class and compute the MAUVE scores between natural instances from each class. We report these metrics in Table 5. We observe that the MAUVE scores are highest when the conditioning and ground-truth labels are aligned across all methods, demonstrating that the label guides the generation effectively. We observe that our approach is more consistently effective at class-conditional generation, particularly for the two most similar classes, business and sci/tech. The GPT-2 baseline is again more susceptible to memorization than our approach.

## 5.4 Sequence-to-Sequence Language Generation

**Baselines.** We compare against directly fine-tuning BART-base and FLAN-T5-base on the XSum summarization and QQP paraphrasing datasets. For diffusion baselines, we compare against the following continuous diffusion models learned in the space of word embeddings: DiffuSeq [16], CDCD [13], DINOISER [73], and GENIE [38]. We also compare against the following discrete diffusion models which learn to invert discrete corruption processes (e.g. masking): Reparameterized Discrete Diffusion (RDM) [76] and DiffusionBERT [19]. We compare directly against the metrics reported in prior work on our datasets. For XSum, we additionally train a DiffuSeq model using the official implementation. We note that Gong et al. [16] typically train their models much longer than ours. The XSum DiffuSeq model, for instance, is trained for over 3 × more epochs than our approach.

For machine translation, we compare directly against the prior work that reported SacreBLEU scores to ensure a fair comparison [48].

Table 6: Seq2Seq Evaluation on QQP. Results from fine-tuned language models are in gray.

| Method | Sampling | Rouge-1/2/L ↑ | BERTScore ↑ |
|---|---|---|---|
| DiffuSeq [16] | Random | 55.2/29.2/52.7 | 82.4 |
| RDM-absorbing [76] | Random | —/—/57.9 | 83.7 |
| RDM-multinomial [76] | Random | —/—/57.3 | 83.7 |
| LD4LG (BART-base) | Random | **62.6/39.0/60.3** | **85.8** |
| LD4LG (FLAN-T5-Base) | Random | 62.1/38.4/59.7 | **85.8** |
| DiffuSeq [16] | MBR-10 | —/—/58.8 | 83.7 |
| RDM-absorbing [76] | MBR-10 | —/—/59.5 | 84.7 |
| RDM-multinomial [76] | MBR-10 | —/—/58.5 | 84.7 |
| DiffusionBERT [19] | MBR-10 | —/—/58.9 | — |
| LD4LG (BART-base) | MBR-5 | **63.3/40.3/61.1** | **86.2** |
| LD4LG (FLAN-T5-Base) | MBR-5 | 63.0/39.7/60.7 | 86.1 |
| DiffuSeq [16] | Oracle-5 | 67.4/43.9/65.8 | 83.7 |
| LD4LG (BART-base) | Oracle-5 | **68.0/46.6/66.0** | **87.2** |
| LD4LG (FLAN-T5-Base) | Oracle-5 | 67.8/46.0/65.7 | **87.2** |
| BART-Base | Nucleus | 51.5/28.1/48.3 | 79.9 |
| FLAN-T5-Base | Nucleus | 55.0/30.1/52.3 | 83.2 |
| BART-Base | Beam | 61.9/39.0/59.5 | 85.5 |
| FLAN-T5-Base | Beam | 63.0/40.1/60.5 | 86.2 |

Table 7: Seq2Seq Evaluation on XSum. Results from fine-tuned language models are in gray.

| Method | Sampling | Rouge-1/2/L ↑ | BERTScore ↑ |
|---|---|---|---|
| DiffuSeq [16] | Random | 18.9/1.3/13.6 | 46.8 |
| LD4LG (BART-base) | Random | 37.6/15.5/30.8 | 74.1 |
| LD4LG (FLAN-T5-Base) | Random | **38.1/15.9/31.2** | **74.8** |
| DiffuSeq [16] | MBR-5 | 19.3/1.7/14.1 | 46.9 |
| LD4LG (BART-base) | MBR-5 | 38.2/16.2/31.5 | 74.5 |
| LD4LG (FLAN-T5-Base) | MBR-5 | **38.7/16.6/31.9** | **75.2** |
| DiffuSeq [16] | Oracle-5 | 23.5/2.3/18.6 | 47.9 |
| GENIE [38] | Oracle-5 | 37.3/15.3/29.4 | — |
| GENIE w/ pre-training [38] | Oracle-5 | 41.2/19.1/33.4 | — |
| LD4LG (BART-base) | Oracle-5 | 42.4/19.4/36.4 | 75.3 |
| LD4LG (FLAN-T5-Base) | Oracle-5 | **43.0/20.0/37.2** | **76.1** |
| BART-Base | Nucleus | 35.1/13.3/27.7 | 73.1 |
| FLAN-T5-Base | Nucleus | 34.6/12.9/27.2 | 72.7 |
| BART-Base | Beam | 39.9/18.0/32.6 | 75.6 |
| FLAN-T5-Base | Beam | 39.7/17.7/32.3 | 75.3 |

**Results.** We present our comparison on QQP and XSum in Table 6 and Table 7. Our approach significantly outperforms recent diffusion language models across both datasets, especially for the more challenging XSum dataset. For instance, DiffuSeq is reasonably effective for QQP, but it struggles with XSum and fails to generate coherent text (see samples in appendix). Our method, on the other hand, is competitive with fine-tuning. LD4LG narrowly outperforms fine-tuning on QQP with MBR decoding, but the fine-tuned models are slightly more effective on the XSum dataset. Across both datasets, LD4LG with oracle sampling outperforms all approaches (including direct fine-tuning methods) with just 5 random samples. This demonstrates that LD4LG has good coverage,
but MBR decoding does not consistently identify the best candidate. In our experiments, we use classifier-free guidance with guidance strength $w = 2.0$. We ablate this choice with validation samples in Figure 3 and observe that such guidance meaningfully improves performance.

Table 8: Machine translation results on WMT14-En-De. Baseline results are from [13, 73].

| Method | Sampling | SacreBLEU | |
|---|---|---|---|
| | | En→De | De→En |
| CDCD [13] | Random | 19.3 | 24.9 |
| LD4LG (MT5-base) | Random | 21.4 | 26.2 |
| Diffusion-LM [36] | MBR-5 | 15.3 | 17.3 |
| CDCD [13] | MBR-10 | 19.7 | 25.4 |
| DINOISER [73] | MBR-5 | 24.3 | 28.8 |
| LD4LG (MT5-base) | MBR-5 | 22.4 | 27.0 |

We report our machine translation results in Table 8. We observe that LD4LG outperforms the Diffusion-LM and CDCD baselines although it lags behind the DINOISER baseline. This demonstrates that our method can effectively take advantage of strong pre-trained multilingual language models for effective multilingual generation.

## 6 Future Work

Our experiments demonstrate that latent language diffusion models can generate high-quality natural language in a variety of settings. In continuous domains, diffusion models are remarkably effective for applications ranging from image editing [41] to solving inverse problems [64]. We are excited to explore the potential applications enabled by effective language diffusion models. We expect that LD4LG is a natural fit for applications such as language editing (e.g. style transfer) and controllable generation (e.g. mitigating toxicity).

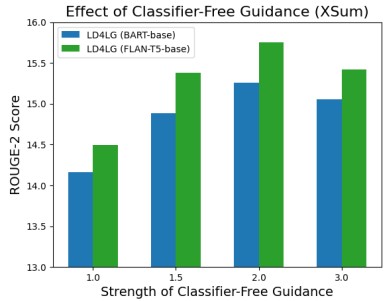

Figure 3: Ablation of classifier-free guidance on the XSum summarization benchmark.

Despite achieving good performance, LD4LG has important limitations. Sampling from diffusion models is slow due to the iterative generative process. LD4LG improves upon some prior continuous text diffusion models (that use 2000 steps) and only uses 250 sampling steps. However, speeding up the inference process of diffusion models is an active area of research and techniques developed for image diffusion can likely be adapted for LD4LG [65, 57]. Song et al. [65], for instance, distilled a trained image diffusion model to produce high-quality samples in a single step. We leave the extension of such techniques to language generation as future work. In subsection 5.4, we observe that diffusion models have excellent coverage but MBR decoding fails to identify the best candidate; developing improved sampling procedures or candidate re-ranking methods would likely improve performance for tasks such as summarization and machine translation.

## 7 Related Work

**Diffusion models.** Diffusion models [61, 22] are a class of generative models that have led to impressive results in image synthesis, recently surpassing Generative Adversarial Networks [17, 12]. These models typically operate directly in pixel-space, learning a distribution over images. Rombach et al. [52] introduced latent diffusion for image synthesis and demonstrated that they can be learned in the latent space of a pretrained autoencoder. Latent diffusion has since been successful in other domains such as audio synthesis [32], symbolic music generation [58], and molecule generation [71].

**Diffusion for Language.** Prior work has focused on directly modeling discrete data by designing diffusion processes for discrete state spaces [25, 3, 26]. Li et al. [36] train a continuous diffusion model in the space of token embeddings that are learned jointly with the denoising objective and decode generations with a rounding step. Strudel et al. [67] scaled up this approach and instead learn the diffusion model in the space of pretrained word embeddings and find that low-dimensional embeddings are better suited for diffusion. Gong et al. [16] extend Diffusion-LM [36] to sequence-to-sequence tasks by concatenating the source and target sequence and only performing diffusion for the target sequence. Chen et al. [8] map words to arbitrary binary strings, represented as a sequence of real numbers. They then train a continuous diffusion model and round the generated sequences to produce binary strings. The authors also introduce self-conditioning, which we adopt for our method.

## 8 Conclusion

In this work, we demonstrate that latent diffusion is an effective paradigm for language generation. To achieve this, we introduce a method for compressing the high-dimensional, variable-length language representations from pre-trained language models into a compact, fixed-size latent representation that can be decoded into natural language. This compact latent representation is, by design, well-suited for learning continuous latent diffusion models. Our latent language diffusion models are effective for unconditional, class-conditional, and sequence-to-sequence language generation. They offer some benefits over fine-tuned auto-regressive language models and significantly outperform recent diffusion language models across a variety of datasets.

## Acknowledgements

This research is supported by grants from the National Science Foundation NSF (IIS-2107161, and IIS-1724282, HDR-2118310). The Cornell Center for Materials Research with funding from the NSF MRSEC program (DMR-1719875), DARPA, arXiv, LinkedIn, and the New York Presbyterian Hospital.

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

# A  Diffusion Models

We present a formal description of diffusion [22, 62, 30]. Diffusion models are latent variable models with latents $\mathbf{z} = \{\mathbf{z}_t | t \in [0, 1]\}$ that are given by the forward diffusion process $q(\mathbf{z}|\mathbf{x})$, with the data, $\mathbf{x} \sim p(\mathbf{x})$, being drawn from an unknown distribution.

The forward process is a Markovian process that iteratively adds Gaussian noise to the data over time

$$q(\mathbf{z}_t|\mathbf{x}) = \mathcal{N}(\mathbf{z}_t; \sqrt{\alpha_t}\mathbf{x}, (1 - \alpha_t)\mathbf{I}), \quad q(\mathbf{z}_t|\mathbf{z}_s) = \mathcal{N}(\mathbf{z}_t; \sqrt{\alpha_{t|s}}\mathbf{z}_s, (1 - \alpha_{t|s})\mathbf{I})$$

where $\alpha_{t|s} = \alpha_t / \alpha_s$ and $0 \leq s < t \leq 1$. The noise schedule, specified by $\alpha_t \in [0, 1]$, decreases with $t$ until the final latent becomes approximately Gaussian, $q(\mathbf{z}_1) \approx \mathcal{N}(\mathbf{z}_1; \mathbf{0}, \mathbf{I})$ — independent of the original data. The forward process therefore defines a transition from the data distribution to a Gaussian distribution.

Given access to the original data $\mathbf{x}$, the forward process can be inverted analytically. For $t > s$, we have

$$q(\mathbf{z}_s|\mathbf{z}_t, \mathbf{x}) = \mathcal{N}(\boldsymbol{\mu}_Q(\mathbf{z}_t, \mathbf{x}, s, t), \sigma_Q^2(s, t)\mathbf{I})$$

where

$$\boldsymbol{\mu}_Q(\mathbf{z}_t, \mathbf{x}, s, t) = \frac{\sqrt{\alpha_s}(1 - \alpha_{t|s})}{1 - \alpha_t}\mathbf{x} + \frac{\sqrt{\alpha_{t|s}}(1 - \alpha_s)}{1 - \alpha_t}\mathbf{z}_t, \quad \sigma_Q^2(s, t) = \frac{(1 - \alpha_s)(1 - \alpha_{t|s})}{1 - \alpha_t}.$$

We utilize this to define our generative process. Because $\mathbf{x}$ is unavailable during generation, we train a neural network to approximate the original data given some noisy latent and the timestep, $\hat{\mathbf{x}}_\theta(\mathbf{z}_t, t) \approx \mathbf{x}$. The denoising network is trained utilizing a regression loss

$$\mathcal{L}(\theta) = \mathbb{E}_{t, \mathbf{x}, \epsilon}[\lambda_t \|\hat{\mathbf{x}}_\theta(\sqrt{\alpha_t}\mathbf{x} + \sqrt{1 - \alpha_t}\epsilon, t) - \mathbf{x}\|_2^2]$$

with some time-dependent weighting $\lambda_t$. This loss function can be motivated as the weighted variational lower bound of the log likelihood of the data under the forward diffusion process [22, 31]. In practice, the denoising network is often parameterized as an $\epsilon$-prediction network [22] or a $\mathbf{v}$-prediction network [57] where the velocity, $\mathbf{v}$, is defined as $\mathbf{v} = \sqrt{\alpha_t}\epsilon - \sqrt{1 - \alpha_t}\mathbf{x}$. These parameterizations can be interpreted as different weighting functions, $\lambda_t$, for the regression objective [57]. We adopt the $\mathbf{v}$-parameterization throughout this work.

With a trained denoising network, we define our generative process as

$$p_\theta(\mathbf{z}_s|\mathbf{z}_t) = \mathcal{N}(\mathbf{z}_s; \boldsymbol{\mu}_\theta(\mathbf{z}_t, s, t), \sigma^2(s, t)\mathbf{I})$$

where

$$\boldsymbol{\mu}_\theta(\mathbf{z}_t, s, t) = \boldsymbol{\mu}_Q(\mathbf{z}_t, \hat{\mathbf{x}}_\theta(\mathbf{z}_t, t), s, t), \quad \sigma^2(s, t) = 1 - \alpha_{t|s}.$$

We therefore substitute our estimate of the clean data into the posterior distribution of $q(\mathbf{z}_s|\mathbf{z}_t, \mathbf{x})$ to parameterize the mean of our generative process $p_\theta(\mathbf{z}_s|\mathbf{z}_t)$. We follow Ho et al. [22] and set the variance of $p_\theta(\mathbf{z}_s|\mathbf{z}_t)$ to $\sigma^2(s, t) = 1 - \alpha_{t|s}$, a choice given by the variance of the forward process.

For generation, we utilize the standard DDPM sampler, also known as the ancestral sampler [22]. We sample some initial noise $\mathbf{z}_{t_1} = \mathbf{z}_1 \sim \mathcal{N}(\mathbf{0}, \mathbf{I})$ and iteratively apply the update rule

$$\mathbf{z}_{t_{i+1}} = \boldsymbol{\mu}_\theta(\mathbf{z}_{t_i}, t_{i+1}, t_i) + \sigma(t_{i+1}, t_i)\epsilon$$

where $\epsilon \sim \mathcal{N}(\mathbf{0}, \mathbf{I})$ and the intermediate timesteps $1 = t_1 > t_2 > ... > t_T = 0$ linearly interpolate between 1 and 0. We use $T = 250$ sampling timesteps by default.

# B  Additional Language Autoencoder Results

We present results for our language autoencoders on XSum, QQP and WMT14 in Table 9. We observe that our proposed language autoencoders are similarly effective for these datasets.

We also ablate the performance as we vary the dimensionality of the latent space in Table 10. We observe, as expected, that the reconstruction performance improves as the dimensionality of the latent space increases and degrades as we decrease the size of the latent representation. We found our default dimensionality of $32 \times 64$ to be generally effective for high quality reconstructions across datasets.

Table 9: Effectiveness of Language Autoencoder

| Method | Latent Dimensions | Hidden Units | XSum | | QQP | |
|---|---|---|---|---|---|---|
| | | | Rouge-1/2/L | BLEU | Rouge-1/2/L | BLEU |
| BART-Base | $L \times 768$ | $\leq 49{,}152$ | 99.9/99.9/99.9 | 99.9 | 99.9/99.9/99.9 | 99.8 |
| BART-Base Autoencoder | $32 \times 64$ | 2048 | 99.8/99.6/99.8 | 99.3 | 99.9/99.8/99.9 | 99.1 |
| FLAN-T5-Base | $L \times 768$ | $\leq 49{,}152$ | 65.7/51.2/59.9 | 45.7 | 26.9/14.1/24.3 | 10.8 |
| FLAN-T5-Base Autoencoder | $32 \times 64$ | 2048 | 99.6/99.3/99.6 | 98.8 | 99.8/99.5/99.7 | 98.5 |
| | | | WMT14 English | | WMT14 German | |
| | | | Rouge-1/2/L | BLEU | Rouge-1/2/L | BLEU |
| MT5-Base Autoencoder | $32 \times 64$ | 2048 | 99.7/99.2/99.7 | 99.2 | 99.8/99.4/99.8 | 99.1 |

Table 10: Ablation of Autoencoder Latent Dimensionality

| Method | Latent Dimensions | Hidden Units | RocStories | |
|---|---|---|---|---|
| | | | Rouge-L | BLEU |
| BART-Base | $L \times 768$ | $\leq 49{,}152$ | 98.8 | 97.5 |
| BART-Base Autoencoder | $32 \times 32$ | 1024 | 97.0 | 92.4 |
| | $32 \times 64$ | 2048 | 99.2 | 97.6 |
| | $64 \times 64$ | 4096 | 99.2 | 97.7 |

# C  Impact of Sampling Steps

We present the results from different sampling configurations for the ROCStories dataset in Table 11. We also report the wall clock time needed to generate the 1000 samples across the different numbers of sampling timesteps while batching the generations with a batch size of 128.

We find that the number of sampling steps introduces a tradeoff between the diversity and the quality of the text, with more sampling steps leading to more fluent but less diverse text and fewer sampling steps leading to less fluent but more diverse text. When using BART-base, the MAUVE score is maximized when utilizing only 100-250 steps, demonstrating that it achieves a reasonable balance between diversity and quality. When utilizing FLAN-T5-base, on the other hand, we find that the MAUVE score improves monotonically with increased sampling steps. This suggests that the latent distribution of the FLAN-T5-base autoencoder may be more challenging to learn. Increasing the capacity of the denoising network or the language autoencoder may therefore be beneficial when using FLAN-T5-base.

We observe that the sampling time scales with the number of sampling steps as expected, although there is also a fixed cost from the reconstruction network and the autoregressive decoder that is independent of the number of sampling steps.

Table 11: Evaluation of different sampling configurations. We use 250 steps by default.

| | Sampling Steps | MAUVE ↑ | Ppl ↓ | Div ↑ | Mem ↓ | Wall Clock Time (1000 samples) |
|---|---|---|---|---|---|---|
| | | | | ROCStories | | |
| Reference | - | $.951_{.007}$ | $21.1_{.3}$ | $.414_{.003}$ | $.362_{.003}$ | - |
| LD4LG (BART-base) | 50 | $.684_{.031}$ | $52.6_{.3}$ | $.407_{.004}$ | $.337_{.003}$ | 1m27s |
| | 100 | $.719_{.022}$ | $38.5_{.8}$ | $.368_{.002}$ | $.392_{.001}$ | 1m55s |
| | 250 | $.716_{.019}$ | $30.6_{.5}$ | $.331_{.005}$ | $.441_{.004}$ | 3m20s |
| | 500 | $.704_{.033}$ | $28.1_{.3}$ | $.313_{.003}$ | $.462_{.003}$ | 5m44s |
| | 1000 | $.667_{.026}$ | $25.9_{.1}$ | $.295_{.002}$ | $.481_{.004}$ | 10m30s |
| LD4LG (FLAN-T5-base) | 50 | $.331_{.028}$ | $67.9_{.7}$ | $.456_{.001}$ | $.283_{.001}$ | 1m34s |
| | 100 | $.421_{.012}$ | $48.7_{.7}$ | $.423_{.002}$ | $.334_{.002}$ | 2m02s |
| | 250 | $.481_{.007}$ | $37.5_{.4}$ | $.389_{.002}$ | $.387_{.002}$ | 3m29s |
| | 500 | $.495_{.024}$ | $32.8_{.6}$ | $.370_{.006}$ | $.413_{.006}$ | 5m51s |
| | 1000 | $.522_{.023}$ | $30.6_{.3}$ | $.360_{.004}$ | $.432_{.005}$ | 10m38s |

Table 12: Evaluation of different nucleus sampling configurations.

|  | ROCStories | | | |
|---|---|---|---|---|
| Sampling Parameter ($p$) | MAUVE $\uparrow$ | Ppl $\downarrow$ | Div $\uparrow$ | Mem $\downarrow$ |
| .90 | $.762_{.027}$ | $19.6_{.3}$ | $.362_{.008}$ | $.718_{.006}$ |
| GPT-2-Medium    .95 | $.788_{.025}$ | $20.0_{.2}$ | $.372_{.002}$ | $.688_{.006}$ |
| .98 | $.782_{.020}$ | $20.2_{.3}$ | $.378_{.002}$ | $.666_{.008}$ |
| 1.00 | $.793_{.024}$ | $20.5_{.4}$ | $.385_{.004}$ | $.637_{.006}$ |

## D  GPT-2 Sampling Ablation

We report an ablation of the nucleus sampling parameter, $p$, in Table 12. The memorization does exhibit some sensitivity to the nucleus sampling parameter, but the memorization is consistently higher than the LD4LG models across all sampling configurations.

## E  Implementation Details

All of the models presented in this work are trained on a single Nvidia A6000 except for the DiffuSeq XSum baseline which was trained with two Nvidia A6000s.

### E.1  Language Autoencoders

We adopt the pre-LN design [70] for both the compression and reconstruction networks and therefore apply layer normalization before all attention and feedfoward blocks. We also adopt query-key normalization [11] and apply RMSNorm [74] to the queries and keys before computing the dot product similarities in the attention mechanism. We found that this enabled training with a larger learning rate which accelerated training. We present the hyperparameters for our language autoencoders across all datasets in this work in Table 13.

We also report additional details such as the number of trainable parameters and training time. The training time is similar across datasets because we use the same hyperparameters, so we simply report the training times for the ROCStories dataset for the monolingual models. For the MT5-base base autoencoder, we report the training time for the German autoencoder which is similar to the English autoencoder. We note that our implementation is not optimized for runtime and that pre-computing and caching the language encoder representations would significantly accelerate training.

### E.2  Latent Diffusion For Language Generation

We present the training details across the different datasets in Table 14. We tuned hyperparameters using the validation MAUVE scores for the ROCStories dataset and found that they generally transferred well across datasets. We therefore used the same hyperparameters across datasets, except that we utilized the L1 loss instead of the L2 loss for the Seq2Seq tasks. Consistent with prior work on image-to-image diffusion models [53], we observed that the L1 loss improved the fidelity of the generations at the cost of sacrificing some diversity. This improved fidelity translated to improvements in our metrics of interest, although the L2 loss may still be desireable for settings where diversity is of greater importance. For the unconditional and class-conditional language models, we did not observe overfitting to be a problem and simply use the final checkpoint for evaluation. For the monolingual Seq2Seq tasks, we utilize the checkpoint with the best validation ROUGE-L. For machine translation, we utilize the checkpoint with the best validation SacreBLEU.

For the machine translation experiments, we observed benefits from rescaling the noise schedule to emphasize training at higher levels of noise. This idea was introduced by Hoogeboom et al. [27] and Chen [7] to improve high-resolution image diffusion models. Both Hoogeboom et al. [27] and Chen [7] shift an existing noise schedule by some scale factor, $s$, to increase the time spent at higher noise levels. Given a noise schedule $\alpha_t$ with SNR $\lambda_t = \frac{\alpha_t^2}{1-\alpha_t^2}$, the shifted noise schedule, $\alpha_{t,s} \in [0,1]$, is

Table 13: Training details for our language autoencoders.

| | Language Model | | |
| --- | --- | --- | --- |
| | BART-base | FLAN-T5-base | MT5-Base |
| Trainable Params | 26M | 26M | 591M |
| Compression Architecture | | Perceiver Resampler [2] | |
| Perceiver Layers | 3 | 3 | 1 |
| Perceiver Dimension | 768 | 768 | 768 |
| Self-Attention Heads | 12 | 12 | 12 |
| Autoencoder Latent Length ($\ell$) | 32 | 32 | 32 |
| Autoencoder Dimension ($d_{ae}$) | 64 | 64 | 64 |
| Reconstruction Architecture | | Transformer [69] | |
| Transformer Layers | 3 | 3 | 1 |
| Transformer Dimension | 768 | 768 | 768 |
| Self-Attention Heads | 12 | 12 | 12 |
| Activation Function | | GELU [20] | |
| Max Seq Length | 64 | 64 | 128 |
| Optimizer | | AdamW [39] | |
| Learning Rate | 1e-4 | 1e-4 | 1e-4 |
| $(\beta_1, \beta_2)$ | (0.9, 0.999) | (0.9, 0.999) | (0.9, 0.999) |
| Batch Size | 256 | 256 | 128 |
| Warmup Steps | 1000 | 1000 | 1000 |
| Learning Rate Schedule | | Linear Decay | |
| Weight Decay | 1e-2 | 1e-2 | 1e-2 |
| Gradient Clipping | 1.0 | 1.0 | 1.0 |
| Training Steps | 50k | 50k | 50k |
| Training Time | 12h38m | 20h17m | 20h29m |

defined so that

$$\frac{\alpha_{t,s}^2}{1 - \alpha_{t,s}^2} = \lambda_{t,s} = \lambda_t * s^2 = \frac{\alpha_t^2}{1 - \alpha_t^2} * s^2.$$

Given $\alpha_t$ and the scale factor $s$, the scaled noise schedule $\alpha_{t,s}$ can be computed in closed-form. Using the relationship that $\alpha_t^2 = \text{sigmoid}(\log(\lambda_t))$ (see Kingma et al. [30]), the new noise schedule can be computed as

$$\alpha_{t,s}^2 = \text{sigmoid}(\log(\lambda_{t,s})) = \text{sigmoid}(\log(\lambda_t * s^2)) = \text{sigmoid}(\log(\lambda_t) + 2\log(s)).$$

We employ a shifted cosine noise schedule with $s = 0.1$ for machine translation. Past work on text diffusion for machine translation observed that training at higher levels of noise improves the models utilization of the conditioning information (i.e. the source sentence) [73].

During the inference process, image diffusion models typically rescale the estimate of the data to the range of pixel values (i.e. [-1,1]) at each sampling step. When we restrict the latent space so that $\|\mathbf{x}_i\|_2^2 = d_{ae}$, we similarly rescale the intermediate estimates of the data to enforce this constraint. This design decision is not critical and similar performance is achieved without this rescaling. We did, however, observe that this made the generative process more robust to large guidance weights which may be important in some settings. This observation is consistent with prior findings from text-to-image diffusion [55].

We also report the wall clock times for training the models, although our implementation could be further optimized to improve training times. The primary cause of the slowdown for AG News compared to ROCStories, for instance, stems from additional validation sampling and logging for class-conditional generation during training.

When decoding the sampled latent vectors, we utilize beam search with a beam size of $4$, a repetition penalty of $1.2$ [29], and prevent generations of duplicate trigrams.

### E.3 BART-Diffusion

For our BART-Diffusion baseline, we utilize the same denoising architecture as our LD4LG method. As discussed in the main paper, the sequence length of the BART features vary with the length of

the input text. During training, the sequence length is simply determined by the training instance. To select the length of the Gaussian noise during generation, we sample a length from the empirical distribution of lengths in the training set.

We observed that the **v**-prediction parameterization was less effective in this setting and the $\epsilon$-prediction parameterization was unstable. We therefore adopted the **x**-prediction parameterization. This is consistent with past work that has found the **x**-prediction parameterization to be more effective for high-dimensional data [36, 8].

Another challenge is that we can no longer control the scale of the latent space. We therefore follow common practices from latent image diffusion and normalize the latent space to have unit variance [51]. When normalizing the latent space, we utilize the first batch of training data to compute the mean for each feature dimension, averaging across the samples in the batch and the sequence lengths of the samples. Therefore, we compute the mean vector $\hat{\boldsymbol{\mu}} = \frac{1}{b\ell} \sum_{b,\ell} \mathbf{x}_{b,\ell}, \hat{\boldsymbol{\mu}} \in \mathbb{R}^d$ where $\mathbf{x} \in \mathbb{R}^{b \times \ell \times d}$ is some batched data. We then compute the global variance across all dimensions in the centered latent space $\hat{\sigma}^2 = \frac{1}{b\ell d} \sum_{b,\ell,d} (\mathbf{x}_{b,\ell,d} - \hat{\boldsymbol{\mu}}_d)^2, \hat{\sigma}^2 \in \mathbb{R}$ to rescale the latent space to have unit variance. We otherwise train this baseline with the same hyperparameters as LD4LG.

### E.4 Diffusion LM

We train our Diffusion-LM models utilizing the public implementation by Li et al. [36][2]. We utilize the provided command and hyperparameter settings for the ROCStories dataset. To adapt it to the AG News dataset, we increase the batch size from 64 to 128 and set the number of training steps to 250k match our training configuration. We otherwise utilize the same hyperparameter settings as the ROCStories model. We attempted to double the learning rate from 1e-4 to 2e-4 to account for the doubled batch size, but observed training instabilities and therefore used the original learning rate of 1e-4.

### E.5 GPT-2

We present the default hyperparameters for the GPT-2-Medium baseline in Table 15. For sampling from GPT-2, we prompt it with a BOS token and utilize nucleus sampling ($p = 0.95$). We use the same repetition penalty of 1.2 [29] that we use for the LD4LG language decoders and similarly prevent generations of duplicate trigrams.

### E.6 DiffuSeq

For the QQP dataset, we compute the metrics with the model generations released by Gong et al. [16]. We utilize the official implementation from Gong et al. [16][3] to train a DiffuSeq model on the XSum dataset. In their work, the DiffuSeq models were trained with the same hyperparameters across all datasets considered, except for the number of training steps which varied across datasets. We therefore adopt their default hyperparameters for the XSum dataset.

We observed that the DiffuSeq models were trained for much longer than our models. The official implementation also utilized gradient accumulation with microbatches of 128 to achieve a large effective batch size of 4096[4]. We trained the XSum DiffuSeq model for 960k iterations which is significantly longer than the 250k iterations used by our LD4LG XSum model. Due to the use of gradient accumulation, this corresponds to 30k gradient updates. The XSum DiffuSeq baseline was therefore trained for over $3.8\times$ more epochs than our method.

A limitation of the DiffuSeq model compared to LD4LG is that it concatenates the source and target sequences as the input to their transformer model. DiffuSeq therefore scales quadratically with respect to the combined length of the source and target sequence. Our denoising network, on the other hand, operates upon a fixed sequence length of $\ell = 32$ latents and only cross-attends to the

---

[2]`https://github.com/XiangLi1999/Diffusion-LM`

[3]`https://github.com/Shark-NLP/DiffuSeq`

[4]We note that the original DiffuSeq implementation had a bug in its implementation of distributed training (see `https://github.com/Shark-NLP/DiffuSeq/issues/37`. We describe the behavior of the original implementation.

source representations. As a result, our method scales linearly with respect to the length of the source sequence[5]. This enables LD4LG to more efficiently incorporate long contexts than DiffuSeq.

By default, the official DiffuSeq implementation limits the combined length of the source and target sequences to a maximum length of 128. This could put it at a disadvantage compared to our model which incorporates up to 256 tokens of the source sequence. To ensure a fair comparison, we also experimented with increasing the maximum sequence length for the DiffuSeq model to 256 tokens, which significantly increases the training overhead. After training the model for 640k iterations, which took 5 days with two Nvidia A6000 GPUs, we observed worse performance than the model using the default length of 128.

### E.7 Encoder-Decoder Language models

We report training hyperparameters for fine-tuning the pre-trained encoder-decoder language models models on the Seq2Seq datasets in Table 16. We perform early stopping with the validation ROUGE-L.

### E.8 Evaluation Metrics

For the MAUVE, ROUGE, BLEU, BERTScore, Perplexity, and SacreBLEU metrics, we utilize the implementations provided by the Huggingface `evaluate` library (`https://huggingface.co/docs/evaluate/`. For SacreBLEU, we follow prior work and use the `intl` tokenizer if the target language is German and use the `13a` tokenizer if the target language is English.

For the n-gram metrics, we utilize the `en_core_web_sm` tokenizer from Spacy (`https://spacy.io/`) to split the generations into tokens.

## F  Dataset Statistics

**ROCStories [42].** The dataset consists of 98,161 instances. We hold out 1,000 instances for validation, 4,000 instances for testing, and utilize the remaining 93,161 instances for training.

**AG News Topic Classification [60].** The dataset consists of titles and short descriptions from news articles. We discard the titles and focus on generating the descriptions in this work. The official train/test splits have 120k training instances and 7,600 testing instances. We hold out 1,000 instances from the training set for validation. We therefore utilize 119k training instances, 1,000 validation instances, and 7,600 test instances.

**XSUM [43].** The dataset consists of BBC articles from 2010 to 2017 covering a wide range of topics (e.g., News, Politics, Sports, etc.). Each example in the dataset consists of a news article and a summary. It has 204,045 training instances, 11,332 validation instances, and 11,334 test instances.

**QQP [9].** The dataset consists of 400k question pairs, where example consists of two similar questions and a binary value indicating whether the two questions have the same meaning. The semantically similar questions can be utilized as a paraphrasing dataset. We use the version released by Gong et al. [16] to enable direct comparison. It has 144,715 training instances, 2,048 validation instances, and 2,500 test instances.

**WMT 2014 English-German [5].** The dataset consists of roughly 4.5 million paired English and German sentences for training. The validation and testing splits each have roughly 3k paired sentences.

## G  Qualitative Examples

We present random unconditional samples from the diffusion models for the ROCStories (Table 17) and AG News (Table 18) datasets. We note that because the Diffusion-LM learns token representations from scratch and cannot model rare words, Li et al. [36] replace rare words with an UNK token. We observe that these tokens are often generated, leading to incoherent text. This problem is particularly

---

[5]For LD4LG, the frozen language encoder still scales quadratically with the source sequence length, but the source representations can be pre-computed and cached prior to training.

Table 14: Training details for LD4LG across different datasets.

| | ROCStories | AG News | XSum | QQP | WMT14-En-De |
|---|---|---|---|---|---|
| Trainable Params | 188M | 190M | 217M | 217M | 218M |
| Sampling Timesteps | | | | 250 | |
| Noise Schedule | Cosine | Cosine | Cosine | Cosine | Shifted Cosine ($s = 0.1$) [27, 7] |
| Regression Loss | L2 | L2 | L1 | L1 | L1 |
| Transformer Layers | | | | 12 | |
| Transformer Dimension | | | | 768 | |
| Self-Attention Heads | | | | 12 | |
| Dense Connections [4] | | | | 3 | |
| Activation Function | | | | GeGLU [59] | |
| Optimizer | | | | AdamW [39] | |
| Learning Rate | 2e-4 | 2e-4 | 2e-4 | 2e-4 | 4e-4 |
| $(\beta_1, \beta_2)$ | | | | (0.9, 0.999) | |
| Batch Size | 128 | 128 | 128 | 128 | 512 |
| Warmup Steps | | | | 1000 | |
| Learning Rate Schedule | | | | Cosine Decay | |
| Weight Decay | | | | 1e-6 | |
| Dropout | 0.1 | 0.1 | 0.1 | 0.1 | 0.0 |
| Gradient Clipping | 1.0 | 1.0 | 1.0 | 1.0 | 0.2 |
| EMA Decay | | | | 0.9999 | |
| Training Steps | 250k | 250k | 250k | 250k | 500k |
| Max Seq Length (Source) | n/a | n/a | 256 | 64 | 128 |
| Training Time (BART-base) | 1d 11h | 1d 20h | 2d 22h | 1d 20h | — |
| Training Time (FLAN-T5-base) | 1d 17h | 1d 21h | 4d 2h | 2d 7h | — |
| Training Time (MT5-base) | — | — | — | — | 9d 16h |

Table 15: Training details for our autoregressive baseline across different datasets.

| | ROCStories | AG News |
|---|---|---|
| Model | | GPT-2-Medium |
| Trainable Params | | 355M |
| Max Seq Length | | 64 |
| Optimizer | | AdamW [39] |
| Learning Rate | | 8e-5 |
| $(\beta_1, \beta_2)$ | | (0.9, 0.999) |
| Batch Size | | 32 |
| Warmup Steps | | 500 |
| Learning Rate Schedule | | Linear Decay |
| Weight Decay | | 1e-2 |
| Dropout | | 0.1 |
| Gradient Clipping | | 1.0 |
| Training Steps | | 100k |

Table 16: Training details for our Seq2Seq baselines model.

| | XSum | | QQP | |
|---|---|---|---|---|
| Model | BART-base | FLAN-T5-base | BART-base | FLAN-T5-base |
| Trainable Params | 139M | 220M | 139M | 220M |
| Max Seq Length (Source) | | 256 | | 64 |
| Max Seq Length (Target) | | | 64 | |
| Optimizer | | | AdamW [39] | |
| Learning Rate | 5e-5 | 1e-4 | 5e-5 | 5e-5 |
| $(\beta_1, \beta_2)$ | | | (0.9, 0.999) | |
| Batch Size | | | 32 | |
| Warmup Steps | | | 500 | |
| Learning Rate Schedule | | | Linear Decay | |
| Weight Decay | | | 1e-2 | |
| Gradient Clipping | | | 1.0 | |
| Training Steps | | | 100k | |

Table 17: Random samples from ROCStories dataset.

| LD4LG (BART-base) | LD4LG (FLAN-T5-base) | Diffusion-LM |
| --- | --- | --- |
| After a long line in line, Amy was ready to carry her cart. She asked if she should put the money in a bag. The cashier gave her a quarter and she opened the bag. She was happy to see that she paid for the amount on the line. The checkier checked when she | Emma was playing with her doll doll. She was having a good time when suddenly she slipped! The doll doll shattered in many places! Emma was so upset she cried and cried! Her mother took her home and got her a new band-aid. | Tom was going to eat with friends. But it was stressed out. So He decided to go to the local bar. But when he realized his friend was too much. The police allowed home to pull him home. |
| Barry was a popular high school student. He always got good grades in school. Barry's friends all met up. He arrived at his new job with a big grin. Barry decided he would start the new job as a teacher. | Max wanted to build a tree in his backyard. He researched guides on what kind of plant to plant. He went online and cut trees so he could see one that would cover large. He bought all his supplies and drove to the farm dealership. They had planted a beautiful backyard in his neighborhood. | Rita was about to go out in the UNK. UNK was the UNK and Rita was very nervous. She took out the ball was beginning to UNK. She kicked the ball still and knew she was a good kid. She looked in her shoulder and immediately ran to the sound. |
| Michael had a crush on a girl. He finally had the courage to talk to her. Michael went over to her and she walked down a hallway. They chatted for hours. Michael wished he had never asked another girl. | John and Molly thought it would be fun to go to Europe. They decided to take their little child to go swimming. The child had a wonderful time playing in the waves. They also had ta lot of local food. They were exhausted when they still had to return home. | I bought a new UNK. It was a UNK. My friends asked it for some money. We didn't listen. I was declined. |
| Ed got a chihuahua. It escaped its cage. Ed was able to free the chihuohua. He wanted to keep him so he let it alone. Ed is able to keep the rest out. | Yesterday lulu went to the theme park. To her surprise her phone fell out at the park. She was so disappointed. But thankfully no one was looking for it. She had to walk home as fast as she could to get it. | Todd was walking his dog with his dog. The UNK hit by minutes close to check something out. There was a small UNK and UNK off of the ground. He got to the UNK's house to find UNK UNK. Todd's dog started to listen to the UNK of it. |
| Maria was getting ready for her trip. She wanted a specific bathing suit, and went to the mall. She tried on many different outfits, but none fit. Maria realized she had found a great deal while shopping. She bought herself a nice suit. | Anna had been friends with her family for years, but curious. Later, Anna's mom told her she might be sick after a bad age. Anna broke up with this, and swore that she would not get sick. That night, Anna threw up all over the house | Stacy wanted to learn how to ride a horse. She found a long one near her UNK. She decided to UNK on. Finally she was able to ride a UNK. Stacy was happy to be her own horse. |

pronounced for the AG News dataset which has a more diverse vocabulary and uses many proper nouns such as names that are out-of-vocabulary. We also present random class-conditional samples for the AG News datasets (Table 19 and Table 20) for all of the classes.

We present examples of sequence-to-sequence generations for QQP in Table 21 and XSum in Table 22. While the DiffuSeq generations are somewhat reasonable for the simpler QQP paraphrasing dataset, the model completely fails to produce coherent summaries for the challenging XSum dataset. This is the case even though DiffuSeq is trained for significantly longer than LD4LG and uses $8\times$ as many sampling timesteps.

Table 18: Random samples from AG News dataset. HTML entities are decoded for readability.

| LD4LG (BART-base) | LD4LG (FLAN-T5-base) | Diffusion-LM |
| --- | --- | --- |
| What could have been a decisive role in Disney's merger of a leading media group, but not only it appears to have been. Last night, the founders of the media conglomerate's leading stock management unit, introduced legislation capping the | Sachin Tendulkar has found himself fit to India's batting squad ahead of this weekend's final and final session of the first Test against Bangladesh in East Oval. | UNK UNK UNK UNK UNK - UNK, UNK - UNK UNK UNK de UNK, the UNK UNK UNK, the UNK UNK of a UNK UNK UNK UNK. |
| The startup provider will provide CRM-based services for small and midsize businesses on its offices. | America Online and Ask Jeeves settle over file-swapping technology that could lead to lawsuits against hundreds of online businesses and result in fraud. | A federal grand judge has reached a new $ UNK stake in UNK for the $ 35 billion, UNK leading investors to the UNK. & <FONT face="verdana, MS Sans Serif, arial, helvetica " size="-2 " Washington UNK ; |
| Real Madrid moved to the top of the Bayern Munich's Premier League standings on Wednesday, with Atletico Atletico in charge following a 2-0 draw against Porto. | Reuters - U.S. oil and gas companies will try to develop and develop a new greenhouse gas system over the next two years in a bid to give a more cautious sense of environmental conditions for the economy, a senior US Energy Department official said on Wednesday. | North Korea's UNK UNK and UNK UNK UNK UNK the UNK of UNK, UNK UNK, UNK UNK UNK 6 - 4, 6 - 4 at the $ UNK US UNK today. |
| Reuters - Three more Americans will be able to make cloning cloned research to make medical research and innovation that brings them to the massive victims of tuberculosis vaccine, the British government announced on Friday. | Andre Agassi upset Carlos Moya 6-4, 6-2, 6-3 to reach the Stockholm Trophy for the first time on Sunday and wrapped up his first grand slam title. | LONDON ( Reuters ) - It was UNK but the European team's UNK man's Davis Cup game was upheld in the last week due to their start into the semi - finals, manager said. |
| Gary Sidson wants the Miami Heat step down from the Dallas Mavericks at the end of the offseason after undergoing medical proceedings to relieve him. Sidson told The Associated Press. "Every wife has a choice" to make him head with | Canada's rules for federal audits pose a threat to Canadian companies, some wanting to keep journalists out of their jobs, Sen. Thomas Powell warned during his annual meeting of the Securities and Exchange Commission last summer. | UNK - UNK UNK, the UNK of the UNK UNK has decided to stop UNK to the old UNK : UNK UNK UNK a UNK, of his father UNK of UNK. |

Table 19: Random conditional samples from AG News dataset. HTML entities are decoded for readability.

| LD4LG (BART-base) | | | |
| --- | --- | --- | --- |
| World | Sports | Business | Sci/Tech |
| President Bush's re-election has been a number of central issues of the Middle East, but it appears to have happened. He headed the US intelligence committee's re-election yesterday, prompting end of the | Australia have dropped their suspension for next week's game of the first cricket series against India, but it seems to have emerged. Last night, owners of the International Cricket Council's leading governing body, announced plans for scrapping the | What could have been the final step in Disney's merger of a leading media group, but it's likely to happen. Last year, the longtime media conglomerate's (NYSE: news - research) research and cable empire topped the $10 | What could have been a role in the digital's business of the record industry, but it already makes a message to investors. Last year, founders of the music industry's leading record companies, proposed legislation scrapping the |
| AP - The Bush administration has agreed to change its portfolio of redeploying additional U.S. troops to Iraq to prevent possible deployment of many U.K. troops there, the White House and Democratic lawmakers said Friday. | You go home, sit uniform and work in four times. Every day now the Expos are starting to change, along with many people. I want to get us about the goals and forecasts of their playoff | The internet phone provider will charge fee-based phone calls more closely to keep consumers in their hands. | The upstart will provide satellite-based phone services and more services to help customers manage their Web applications. |
| The government stepped up a programme yesterday to monitor Russia's school siege, including a school in Beslan in a school where more than 400 people have fled Russia. | Real Madrid could be in trouble to disrupt Bayern Munich's Champions League clash with Porto at the San Siro on Sunday. Although the recent hat-trick from Ronaldinho has helped Brazil | The Kremlin asked a court order yesterday to punish Irina Yukos' CEO, chairman of Russia's state gas monopoly and the beleaguered oil firm Yukos. | The Cassini-US spacecraft continues to monitor Saturn's largest moon Titan, Saturn's larger moon Titan. A region where the swirling of dust and dust have triggered Saturn |
| Reuters - Prince Thatcher will be released in Cuba after undergoing a brain surgery, her father said on Friday as she flew to the Middle East to help patients evacuate her. | AP - Terrell Owens will be suspended indefinitely for the Kansas City Chiefs due to Hurricane Frances, and he will return to the team Sunday when they face the Tampa Bay Buccaneers. | Where you've ever seen any wireless lending using the Internet you see in your house, or brings it to a great shift when you teleport to the net? | Reuters - stem cells can be used to make cloned cells for medical research experiments and innovations that could open the open to future women cloning, British researchers reported on Friday. |
| NEW DELHI: Prime Minister Manmohan Singh asked the government to do more at reviving the Kashmir peace process with India. Singh, a spokesman for the Association of Ase Nations on Monday said the Indian Government is willing to post off | ATHENS – Paul Hamm made a surprise exit from the Athens Olympics in favor of the Olympics after entering the Games but plans to give him, his spokesman said Monday. "No one has a mistake" to the US gymnastics body. | MOSCOW (CBS.MW) – Supporters of Russia's beleaguered oil firm Yukos have announced they will file for bankruptcy this week to press ahead with a $1.1-billion back-tax bill. | NewsFactor - IBM (NYSE: IBM) is acquiring its WebSphere division in a deal valued at US $160 million in cash. Meanwhile, the companies said the deal valued of US $1.5 billion for Sybase (Nasdaq: ADABECKs) (Rasdaq |

Table 20: Random conditional samples from AG News dataset. HTML entities are decoded for readability.

| LD4LG (FLAN-T5-Base) | | | |
| --- | --- | --- | --- |
| World | Sports | Business | Sci/Tech |
| British Prime Minister Tony Blair met security officials at Queen's Palace in London just two weeks after the US-led invasion of Iraq. | LeBron James did all of them they needed, leading the Sacramento Kings to a 92-80 victory Friday night against the Toronto Raptors at Air Canada Center. | US Airways has reached an agreement with its pilots representing the nation's biggest airline, just three weeks after emerging from bankruptcy protection. | The first close-up images of Saturn's largest moon Titan have been seen several times or long on August 28th this image captured by the Cassini Space Telescope. |
| AFP-based Southeast Asian countries agreed to negotiations on a job-cutting deal that could result in a trade fiasco in the world's biggest economy and a solution to post-war conflict. | AFP-Ajax Amsterdam coach Rafael Nedved set aside to order a shake-up of the Spanish premier league club after a month of speculation. | SAN FRANCISCO (CBS-MW): Cingular Wireless is close to talks over a possible US $11 billion purchase of US storage firm Veritas Software. | America Online and Ask Jeeves continue to work on file-swapping technology that could lead to hundreds of online businesses of revolution. |
| WASHINGTON (Reuters) - U.S. health and health officials agreed on Wednesday to begin testing a new biodegradable drug that tests questionable risks of heart disease, with a newly published report from the Food and Drug Administration (FDA) only the only one likely to have | AP - The University of Washington announced Wednesday that its new football coach, Tyrone Willingham will be serving as head coach at Notre Dame for its only second run in the vacancy. | WASHINGTON (Reuters) - U.S. chain store sales grew by 0.2 percent in the latest week, held back by the biggest year-earlier pace in history, the Commerce Department said on Wednesday. | Reuters's web search leader Yahoo will begin testing a new desktop search engine with Web searches next month in a bid to maintain Google's remaining foothold in the Internet world, a company executive said on Wednesday. |
| Voters had to cast their ballots in Afghanistan's landmark presidential election with an easy and easy win over Hamid Karzai as its first popularly elected leader. | Miguel Jimenez shot a 4-under 66 to take a one-stroke lead over Carlos Moya after the third round of the PGA Tour tournament. | 'We're all wanted to win one - I all want to do that, "Tiger Woods' had already had a very crucial victory. | Grand Theft Auto needed only 200,000 to get on Xbox; although they just wanted to do so, it's just very impressive. |
| AP - Records of the oil-for-food market and job embargoes produce yet another political stalemate for Democratic Sen. John Kerry's effort to end the lawsuits of journalists, including the ones carried out by the White House, at the summer weekend | AP: The Boston Red Sox are old, but the slugger of baseball is old, despite an expected flurry of diplomacy and outrage from fans that the holidays will bring their way at Fenway night, Sunday night end | The new Securities and Exchange Commission rules will lead to a deep defence of corporate business and the company will continue its oversight of accounting practices, US U.S. Senator Thomas Powell warned during a brief with the Security and Exchange Board last night. | If file-sharing services are legal, the Federal Court of Justice (FTC) is not liable for copyright infringements in the UK, which was brought in by the Federal Trade Commission (FTC) ruling, brought in yesterday. |

Table 21: Samples from QQP Paraphrasing Dataset.

| **Source:** What are some creative and innovative business ideas with less investment in India? | | |
| --- | --- | --- |
| **Reference:** What are some best business ideas with minimum investment? | | |
| LD4LG (BART-Base) | LD4LG (FLAN-T5-base) | DiffuSeq |
| What are some innovative business ideas with less investment? | What are some creative business ideas with lesser investment? | what are best business ideas with less class investment? |
| What are some new business ideas with minimum investment? | What are some new business ideas with minimum investment? | what are some business ideas with less investment in india? |
| What are some new business ideas with lesser investment? | What are some new business ideas with lesser investment? | what are some business ideas with available in india? |
| **Source:** Can height increase after 25? | | |
| **Reference:** Can someone increase their height naturally after 19? | | |
| LD4LG (BART-Base) | LD4LG (FLAN-T5-base) | DiffuSeq |
| Is it possible to increase their height after the age of 25? | Is it possible to increase height after the age of 21? | how can i increase taller after 21? |
| Is it possible to increase the height after the age of 21? | Is it possible to increase height after the age of 25? | how can i increase height at after 21? |
| Is there any way to increase the height after 21? | Is it possible to increase height after age of 21? | how girls is increase our height? can be his, 21 18 years? |

Table 22: Samples from XSum Summarization Dataset. Parts of the articles are omitted for brevity.

**Article:** Two-year-old Lane Thomas Graves had been playing in the sand near the resort's Seven Seas Lagoon when he was dragged underwater by the creature... The lighthouse has been installed near to where the attack occurred... A Disney spokesperson said they hoped the monument would spread awareness for the Lane Thomas Foundation, which also uses the lighthouse as its logo. Who is liable for alligator boy's death? "The lighthouse sculpture has been installed to help spread awareness of the Lane Thomas Foundation, which was established to provide assistance and support to families whose children need organ transplants," Walt Disney World said in a statement.

**Reference:** Walt Disney World has unveiled a lighthouse memorial for a young boy who was killed by an alligator while on holiday at the Florida theme park.

| LD4LG (BART-Base) | LD4LG (FLAN-T5-base) | DiffuSeq |
|---|---|---|
| A giant lighthouse in memory of a US boy who was fatally attacked by an alligator has been installed at Walt Disney World. | A sculpture lighthouse has been installed in memory of an American boy killed by an alligator at a Florida theme resort. | a teenager has been ap graves claimed in after the in the over killing sand of an resort. |
| A national lighthouse in memory of a US boy who died when he was attacked by an alligator has been installed at Disney World. | A US boy has been killed by an alligator while playing in sand at a Florida theme resort, officials say. | a woman has died from s over a a caused parents in county accident a. |
| A giant lighthouse has been installed at Walt Disney World in Florida in memory of a US boy killed by an alligator. | A Disney lighthouse has been installed in memory of an American boy killed by an alligator on a Florida holiday resort. | a speech of man who killed the using his leicester has by a resort been some been has resort playing by raised. |

**Article:** The Sky Blues currently play in Coventry's Ricoh Arena but had a long dispute with the stadium's previous owners... In a statement, Rugby Borough Council said its leader and the council's executive director and head of planning had met with Coventry City in March. "The club requested the meeting to understand how the council would deal with any planning application for potential stadium sites in the borough of Rugby," it said. It said the plans would need to be finalised by September to be included in the council's local plan, but added that a site had yet to be identified. Peter Ward, from Sky Blues Supporters' Consultative Group, said he was pleased to hear that things were "moving" with the club's search for a new home. "It's good that finally there is some evidence things

**Reference:** Planners in Rugby have revealed they have been in talks with Coventry City Football Club about building a stadium in the borough.

| LD4LG (BART-Base) | LD4LG (FLAN-T5-base) | DiffuSeq |
|---|---|---|
| Premiership club Rugby Football Club have met with Coventry City Council to discuss a potential new stadium in the city. | Coventry City have held talks with Rugby Borough Council to consider plans for a new stadium. | coventry city will have midfielder barack their for as has a poor his at side the club one back until. |
| Coventry City's Rugby Football Club have met with local authorities to discuss the potential site of a new stadium for next season. | Coventry City have held talks with Rugby Borough Council to discuss the search for a new stadium. | owners city's david stadium been has council been sky a to league by talks from the club'at london. |
| Coventry City football club has met with Rugby Borough Council to discuss potential sites for a new stadium in the city. | Coventry City FC has held talks with the borough council to discuss a new stadium. | coventry city have set a stadium new league over cup death following a after deal - was league their a club club. |

