# OpenReview forum: "Latent Diffusion for Language Generation"
_NeurIPS.cc/2023/Conference — NeurIPS 2023 poster_

### Official Review · Reviewer_TpnG · 2023-07-02

**Soundness:** 3 good
**Presentation:** 3 good
**Contribution:** 3 good
**Rating:** 6
**Confidence:** 4

**Summary:**

This work proposes a new type of diffusion language model based on latent representations. The proposed approach leverages the continuous representation space produced by pre-trained encoder-decoder language models (BART and FLAN-T5). To embed the variable-length encoder features, the author(s) propose to adopt Perceiver Resampler to construct a fixed length latent representation. After obtaining the diffused representation, a reconstruction network is used whose output is later taken by the frozen language decoder to generate the output.

Extensive experiments are performed on a few types of tasks across different benchmarks. The experimental results verify that the proposed approach outperforms existing diffusion-based methods and performs comparably with directly fine-tuned language models.

Good discussions on future work are included in this work.



**Strengths:**

* The paper is generally well-written and the presentation is clear.
* The proposed model is novel in terms of architectural design and motivations.
* Extensive experiments are conducted to sufficiently evaluate the method and the results are convincing.


**Weaknesses:**

* One major weakness of this work is its potential impact to the research community in the long-term. From the presented results (Table 6), simply fine-tuning the language model still yields the best results across the board. In addition, the existing autoregressive language model such as ChatGPT and GPT-4 have proven their exceptional capabilities in various applications. On the other hand, it is hard to judge the performance of diffusion-based approaches, especially when the model is scaled up. I believe scaling up the proposed model is out-of-scope of the this work, but it worth considered by the author(s).
* **An important comparison is missing**. From the first impression, the encoding of language encoder and the sequential decoding of language decoder are shared by both the proposed method and the standard autoregressive language model. The additional diffusion process would inevitably increase the inference latency. While the author(s) discuss the downside of inference latency in Section 6, it is important to compare both approaches systematically.



**Questions:**

1. Could you elaborate more details of the language autoencoder experiments in Section 5.1? Do you train the compression and reconstruction networks to let the model replicate the input in its output?
2. In line 268, why do you need to specify the length for inference. I think the generation length is determined by the emission of EOS token of the language decoder. Please correct me if I am wrong.

**Limitations:**

Please see the sections of weaknesses and questions.

---

> ### Author Rebuttal · Authors · 2023-08-10
>
> **Re: Long-term Impact**
>
> We appreciate the reviewer raising the important concern about the long-term impact of our approach. The reviewer is correct that autoregressive transformers have continued to advance rapidly, demonstrating exceptional capabilities in a variety of language tasks.
>
> However, we believe language diffusion models provide complementary strengths in areas like sample diversity and controllability. Our latent diffusion approach was more effective at generating novel text and more effective at generating text from a specified topic than GPT-2, an autoregressive model with nearly double the trainable parameters as our latent variable model. For the Seq2Seq tasks, our oracle evaluation demonstrated that drawing just 5 samples from the diffusion language model consistently produced a higher-quality generation than that produced by the fine-tuned language model. This demonstrates that the diffusion model has excellent coverage over the hypothesis space and that further exploration of re-ranking techniques could lead them to consistently outperform the fine-tuned language model.
>
> While we agree that observations made in our compute-regime cannot be extrapolated to that of state-of-the-art GPT models, promising results in modest compute-regimes are critical for motivating serious attempts at scaling. To return to the point of long term-impact, we believe that language diffusion is still in its early days as a paradigm, and further research is needed to realize its full potential. As a point of comparison, diffusion models did not surpass GANs on image synthesis benchmarks until 6 years after their original introduction. They have since become the dominant paradigm in that domain, though whether language diffusion models will follow a similar trajectory remains to be seen.
>
> We hope our work provides a foundation for further research into language diffusion as a promising new paradigm. In this work, we are the first to demonstrate that diffusion models can be integrated with existing pre-trained language models for high quality language generation. We are particularly interested in scaling up hybrid approaches such as the one demonstrated in this work to recent open source autoregressive language models such as LLama.
>
> **Re Auto-encoder training details:**
>
> Yes, we train the auto-encoding modules such that the compact features produced by the compression network are mapped to features that guide the language decoder, through the cross-attention mechanism, to generate the input language. We train the autoencoder with the standard cross-entropy loss to re-generate the input language from the bottleneck representation.
>
> **Re line 268:**
>
> For the BART-diffusion baseline described in 268, we explore learning a diffusion model directly in the feature space produced by the pre-trained BART model. This enables us to explore the benefits of learning our language autoencoder. Because language varies in length, the size of the BART latent diffusion space also varies with the sequence length of the text. For this baseline, a sequence length l must therefore be selected to generate samples. In contrast, our method LD4LG has a latent space of a fixed size. As the reviewer mentioned, LD4LG is therefore able to generate sequences of different lengths by solely relying on the decoder’s emission of the EOS token. The comment about specifying the length for inference only applies to the BART-diffusion baseline.
>
> **Re Inference Latency:**
>
> We appreciate the reviewer raising this important point. They are correct that our method introduces additional overhead compared to directly using the pretrained language models. The reviewer rightly emphasizes the need to quantify the inference latency trade-off through direct comparison to the fine-tuned language models.
>
> One of the strengths of our approach is that it enables the adaptation of advanced techniques for accelerating inference from other continuous domains such as images. There is already a significant body of work focused on reducing the inference time of image diffusion models. Improved sampling algorithms such as DPM-Solver++ [1] also enable the use of fewer timesteps at inference than the traditional DDPM sampler used in our work. To illustrate this, we report the time needed to generate 1000 validation summaries for the fine-tuned language model, FLAN-T5, and our approach on the XSum dataset using the traditional DDPM sampler employed in our original submission as well as the newer DPM-Solver++.
>
> | Method | Sampler | Timesteps | Seconds | ROUGE-L |
> |-|-|-|-|-|
> | FLAN-T5 | Beam | n/a | 26.6 | 31.7 |
> | LD4LG (FLAN-T5) | DPM-Solver++ | 10 | 73.3 | 29.6 |
> | LD4LG (FLAN-T5) | DPM-Solver++ | 15 | 82.5 | 29.9 |
> | LD4LG (FLAN-T5) | DPM-Solver++ | 20 | 90.8 | 30.2 |
> | LD4LG (FLAN-T5) | DDPM | 250 | 547.6 | 30.7 |
>
> While the added latency during generation remains a limitation of our approach, we observe that the recently developed DPM-Solver++ sampler can accelerate our framework by over 5x, although performance does degrade slightly. We are excited about extending other advanced techniques, such as progressive distillation [2] or consistency models [3], which have successfully distilled image diffusion models to perform high-quality generation in a single step models to language.
>
> [1] Lu, Cheng, et al. "Dpm-solver++: Fast solver for guided sampling of diffusion probabilistic models." (2022).
> [2] Salimans and Ho. "Progressive Distillation for Fast Sampling of Diffusion Models."
> [3] Song, Yang, et al. "Consistency models." (2023).

---

> > ### Comment · Reviewer_TpnG · 2023-08-13
> > **Thank you for your comprehensive response**
> >
> > Thank you for the rigorous replies. I appreciate the author(s)' honest acknowledgments of this work's limitations. I have raised my score to 6.

---

### Official Review · Reviewer_LWig · 2023-07-07

**Soundness:** 3 good
**Presentation:** 3 good
**Contribution:** 2 fair
**Rating:** 4
**Confidence:** 5

**Summary:**

The paper proposes a latent diffusion model for text generation, by stacking pre-trained powerful language models and fine-tuned projection layers and diffusion networks together. The input is first encoded into hiddens, then the diffusion model generates the hidden used as the condition of the language decoder. The final results are generated then. Experiments are conducted on both conditional and unconditional language generation tasks.

**Strengths:**

- The paper is well-written and easy to follow.

- The method part is in detail which makes readers understand the model immediately.

**Weaknesses:**

- The contribution is weak.
   1) the diffusion model is only utilized to generate the condition of the LM decoder, which means that the model just provides another projection of the original LLM encoder output. Why is this step necessary? As a cost, the generation speed is very slow. I saw in the appendix that generating 1k samples needs 3 minutes. What will the original BART cost?
   2) The performance of the model heavily depends on the backbone LM. For example, BART outperforms FLAN-T5 by a large margin. What if training from scratch? What is the performance with other architectures? What if the performance of backbones varies from task, for example, in some tasks BART is better but in others FLAN-T5 is better? This kind of dependence strictly limits the generality of the model.

- The experiments lack some new baselines, such as GENIE, Difformer, etc. The choice of datasets is also wired, why not conduct experiments on exactly the same ones as Diffuseq?

**Questions:**

Please answer the questions in Weaknesses.

What are the purpose and benefits of utilizing diffusion in such a framework?

**Limitations:**

The authors have discussed limitations.

---

> ### Author Rebuttal · Authors · 2023-08-10
>
> We discuss our contribution in our overall response.
>
> We would like to emphasize that at inference-time, there is no encoder output. Therefore the diffusion model is not providing a projection of the encoder output, but sampling novel latent codes. The diffusion network iteratively refines a latent code (starting from noise) to model high level text semantics and the decoder produces fluent discrete text conditioned on the latent code.
>
> **Dataset Selection**
>
> We explore unconditional, class-conditional, and Seq2Seq language generation settings, while DiffuSeq focused only on extending Diffusion-LM to Seq2Seq tasks. We thus utilized a wider range of datasets than just those employed by the DiffuSeq paper.
>
> For Seq2Seq tasks, we use the QQP paraphrasing dataset (also used by DiffuSeq) and the XSum summarization dataset. XSum is a well-established seq2seq benchmark widely adopted by the research community, as evidenced by its use across influential works such as BART, FLAN-T5, PaLM, and others. We found the dataset selection in the DiffuSeq paper, beyond QQP, to be non-standard and observed problems with data quality.
>
> For query generation, the DiffuSeq authors use a dataset derived from a QA dataset with string-matching heuristics. Specifically, the context paragraphs are split into sentences and sentences that contains the answer are paired with the original question to form training instances. This often removes critical context needed to answer the question, resulting in incoherent pairings. Here are some representative examples:
>
> | Source Sentence | Target Question |
> |-|-|
> | The year 1921 in science and technology included many events , some of which are listed here | Rudolph Valentinos movie premiere was in which year |
> | The butterfly tattoo is pretty and feminine without being overly flowery or sentimental . | The is no known language without a word for this creature - what |
>
> In these instances, the answer is in the sentence (1921, butterfly), but the context occurred in surrounding sentences.
>
> The DiffuSeq paper also explores dialogue generation and text simplification. The Commonsense Conversation Dataset contains a lot of profanity and sexual content, and is non-standard benchmark. Furthermore, the authors discard the conversation history and use only utterance/response pairs which leads to disjointed instances. Text simplification, while reasonable, lacks the level of adoption as summarization and is related to paraphrasing, which we evaluate.
>
> We aim to provide results on established benchmark datasets that have been widely used and vetted for quality. The datasets used in both DiffuSeq and DiffusionBERT are QQP and QG (the query generation dataset discussed above). See our response to Reviewer SqZp for QQP comparisons. We ran additional experiments with the QG dataset and report results below.  We outperform the baselines on this dataset as well and will add this comparison to the camera-ready. We will also include machine translation results on the WMT14-En-De dataset to further demonstrate the versatility of our approach (see response to reviewer SqZp).
>
> | Method | MBR | ROUGE-L |
> |-|-|-|
> | DiffuSeq | 10 | 36.65 |
> | DiffusionBERT | 10 | 34.20 |
> | LD4LG (bart-base) | 5 | **37.07** |
>
> **Speed**
>
> As the reviewer mentioned, using our approach will be slower than the base LM. However, there is a significant body of work focused on reducing the inference time of image diffusion models that would be applicable to our continuous, latent language diffusion model. For example, Salimans et al. [1] and Song et al. [2] propose methods to distill diffusion models to perform single-step generation. Adapting such techniques to text diffusion will drastically improve the speed of text diffusion models. See our response to Reviewer TpnG for additional timing comparisons.
>
> [1] Salimans and Ho. "Progressive Distillation for Fast Sampling of Diffusion Models."
> [2] Song, Yang, et al. "Consistency models." (2023).
>
> **Backbones**
>
> We thank the reviewer for raising this important point. They are correct that the performance of our approach depends on the backbone LM used. However, this dependence is not unique to our framework - it is also true for standard fine-tuning, prompting, and in-context learning with LMs.
>
> We focus on BART and T5 as they are well-established LMs shown to be generally effective across a diverse range of language tasks in prior work. We do observe that BART outperforms T5 on some datasets and vice versa. This highlights the value of exploring different backbone models to determine the best fit for particular applications and tasks. However, this exploration is also beneficial when fine-tuning LMs for a specific task.
>
> Our LD4LG framework can flexibly incorporate improved base models over time, which is one of the benefits of incorporating pre-trained LMs. Our experiments in response to Reviewer SqZp, for instance, demonstrates that our framework can be extended to multilingual settings by utilizing a multilingual LM.
>
> While we leave investigation of this to future work, we expect training from scratch to result in significantly worse performance compared to leveraging pretrained LMs, as is commonly observed. We thus leverage strong pretrained LMs to provide a high-quality starting point.
>
> **Additional baselines**
>
> We thank the reviewer for the references. GENIE uses the XSum dataset, so we present that comparison here. The GENIE paper uses our oracle setting in their work so we report our oracle performance. We present the results for their method when using the best of 5 to match our evaluation.
>
> | Method | Timesteps | ROUGE-1/2/L |
> |-|-|-|
> | GENIE w/ pre-training (n=5) | 2000 | 41.2/19.1/33.4 |
> | GENIE (Oracle; n=5) | 2000 | 37.3/15.3/29.4 |
> | LD4LG (BART-base) (n=5) | 250 | 42.4/19.4/36.4 |
> | LD4LG (FLAN-T5-base) (n=5) | 250 | **43.0/20.0/37.2** |
>
> We observe that LD4LG significantly outperforms GENIE even when GENIE is pre-trained on a large external corpus.

---

> > ### Author Response · Authors · 2023-08-21
> >
> > In light of our new experiments and favorable comparison against the additional reference you provided, do you have any additional questions or concerns? We will try to respond if we are able in the limited time remaining for the discussion period. Thank you for your valuable comments. We look forward to incorporating your feedback into the final version of the paper!

---

> > ### Comment · Reviewer_LWig · 2023-08-21
> >
> > Thanks a lot for your effort in providing more results! My concerns are mainly about the contribution and experimental comparison: 1) borrowing ldm from cv to nlp is an incremental idea and does not meet the bar of NeurIPS; 2) more experiments with recent text diffusion baselines should be done, i.e., on the same datasets and tasks.
> >
> > I'll hold my original score.

---

> > > ### Author Response · Authors · 2023-08-21
> > >
> > > We would like to further emphasize our extensive experimental comparisons. We list below the various text diffusion methods, along with their publication date, that we have compared against in our original submission and during the rebuttal:
> > >
> > > Diffusion-LM, NeurIPS 2022
> > >
> > > DiffuSeq, ICLR 2023
> > >
> > > DiffusionBERT, ACL 2023
> > >
> > > CDCD, Arxiv December 2022
> > >
> > > GENIE, Arxiv Feb 2023
> > >
> > > DiNOISER, Arxiv Feb 2023
> > >
> > > Reparameterized Discrete Diffusion (RDM), Arxiv Feb 2023
> > >
> > > Given that the submission deadline was in May 2023, we believe that this represents a reasonably comprehensive comparison against recent text diffusion methods. Because text diffusion is in its early stages, the community has not yet settled on an established set of benchmarks and, as discussed, some of the datasets that have been used are nonstandard and have significant quality problems. We compare against the above work when possible and the comparisons are consistently favorable.

---

> > > > ### Author Response · Authors · 2023-08-21
> > > >
> > > > Although latent diffusion is commonly used for images, its extension to language presents unique challenges given the discreteness of the domain. Latent diffusion models are reliant on the strength of the underlying autoencoder; image diffusion models train an autoencoder from scratch, but we introduce a new perceiver-based network to effectively and efficiently adapting existing pre-trained language models into strong language autoencoders. As multiple reviewers acknowledge, we are distinctive from other diffusion language models in developing a hybrid approach utilizing the complementary strengths of diffusion and autoregressive generation.

---

### Official Review · Reviewer_T1gz · 2023-07-07

**Soundness:** 3 good
**Presentation:** 2 fair
**Contribution:** 2 fair
**Rating:** 3
**Confidence:** 4

**Summary:**

The paper demonstrates that encoder-decoder language models can be used to efficiently learn high-quality language autoencoders and to learn continuous diffusion models in the latent space of language autoencoders so that continuous latent representations of natural language can be sampled using pre-trained decoders.

**Strengths:**

1. The paper proposes an additional compression module which maps high-dimensional encoder representations to low-dimensional fixed-length representations. And the corresponding reconstruction network is trained to map these fixed-length features back to high-dimensional features, guiding the frozen language decoder to reconstruct the original language.
2. The paper verifies the effectiveness of the model's unconditional, class-conditional, and sequence-to-sequence language generation methods on several different datasets.

**Weaknesses:**

1.  The motivation of the paper is somewhat weak. Why do we need to apply the latent diffusion model on text generation? What are the advantages over autoregressive and non-autoregressive models? It seems that the performance (MBR-5 in Table 6) is worse than the original AR model. Maybe it is faster? Overall, the lack of clear motivation makes the paper read like a technical report.
2. I have reservations about the ability of the variational paragraph embedder to learn effective representations of sentences with different lengths. What would happen if there is a significant difference in sentence lengths, such as one being very short (e.g., 5 tokens) and the other being much longer (e.g., 512 tokens or more)? Moreover, what is the influence of fixed length number, $l$ in line 102?

**Questions:**

1. Is the autoencoder joint training with the denoising network?
2. What is the influence of different VAE models on the final generation performance?

**Limitations:**

Yes, the authors clearly give the limitations of their work.

---

> ### Author Rebuttal · Authors · 2023-08-10
>
> **Motivation:**
>
> As noted in our overall response, a key motivation of our work is exploring how diffusion models can provide complementary strengths to autoregressive generation for text. Our approach aims to combine the benefits of both methods by learning the diffusion model in the continuous latent space of a pretrained autoencoder.
> As the reviewer rightly asks, the diffusion process is critical for enabling enhanced sampling, controllability, and diversity compared to directly using the discrete token features. By smoothing the features into a continuous latent space, sampling from the prior becomes more flexible and expressive. The autoencoder handles converting the samples back into the discrete text domain. In this way, our hybrid approach benefits from both the diversity and controllability of diffusion as well as the discrete text modeling capabilities of autoregressive decoders. We hope our work helps lay the foundation for realizing the immense potential of diffusion models to advance text generation.
>
> **Sequence length:**
>
> Our framework relies on the autoregressive decoder's capacity for generating sequences of different lengths. If the decoder struggles to model a dataset where the lengths are bimodally distributed, then this is a limitation that will be shared by auto-regressive models and our model. In contrast to other diffusion models that need to model the length of sequences directly, we offload that part to the decoder and our diffusion model just generates samples in the fixed sized latent space.
>
> We develop our language autoencoders across 4 diverse datasets (6 including our machine translation experiments conducted in response to Reviewer SqZp), each with input sequences of varying length, and observe empirically that our proposed autoencoders obtain near perfect reconstruction (i.e. BLEU and ROUGE scores of ~99). The autoregressive component of our language autoencoder is therefore effective at mapping the fixed-length latent representations to variable length natural language utterances with high fidelity.
>
> The reviewer raises a good point that there must be some limit to this sequence length capacity, although we did not encounter it in our experiments. Investigating the use of our framework for open-ended generation that requires maintaining long-term coherence is interesting future work but beyond the current scope.
>
> **Length of the latents:**
>
> For the latent space, our goal was to maximize compression while maintaining strong reconstruction performance with the auto-encoder. This accelerates learning for the diffusion model while maintaining the capacity for high-fidelity language generation. We observed that 32x64 dimensional latents led to high quality reconstructions across all datasets and the reduction in sequence length accelerated model training. When ablating the size of the latents for the auto-encoder, we observed that further decreasing the dimensionality of the latents (e.g. 16x64 or 32x32) began to degrade reconstruction performance. We will add this ablation to the supplementary material.
>
> **Training:**
>
> We follow the practice established by latent image diffusion models and train a high quality auto-encoder prior to learning the diffusion model. We then freeze the autoencoder and train the diffusion network to generate samples from the latent feature space of the frozen autoencoder. We will clarify this in the final version.
>
> **Different Auto-encoder Models:**
>
> In our experiments we utilize both BART and FLAN-T5 to develop language autoencoders. We observed that they performed similarly in most cases. When ablating the dimensionality of the latent space for the autoencoder, we observed strong performance as long as the language reconstruction quality was high. Please see our response to Reviewer LWig (the **Backbones** section) for further discussion on this point.

---

> > ### Comment · Reviewer_T1gz · 2023-08-18
> >
> > Thanks you for the clarifications.
> >
> > The research motivation still remains somewhat unclear to me.
> >
> > The author also did not directly address my question about the generated lengths. The supposed advantages of having both Diffusion and AR simultaneously also bring forth issues. For very short texts, it increases computational overhead. For very long texts (>=512), the effectiveness is also unknown. Although the author mentions that the sentence lengths differ in the various datasets used, without specific statistical results, we are completely unaware of whether the model truly possesses the capability to generate sentences of varying lengths. Lastly, the generalizability of the fixed hidden space size configuration is questionable, and I find it hard to believe that this method could be applicable to different generation tasks through pre-training in the future. Additionally, in Diffusion-LM[1], there are controlled experiments concerning sentence length.
> >
> > The claimed controlled advantages in the paper are compared to a very limited set of models. Diffusion-LM also asserts the advantages of their model in terms of controllability, and the compared models include PPLM[2], FUDGE[3], DELOREAN[4], COLD[5], and GPT-2. However, there is no overlap in the comparison datasets between this article and Diffusion-LM, making it difficult for me to make a judgment.
> >
> > Therefore, I will maintain the score.
> >
> > [1] Diffusion-LM Improves Controllable Text Generation.
> > [2] Plug and play language models: A simple approach to controlled text generation.
> > [3] FUDGE: Controlled text generation with future discriminators.
> > [4] Back to the future: Unsupervised backprop-based decoding for counterfactual and abductive commonsense reasoning.
> > [5] Cold decoding: Energy-based constrained text generation with langevin dynamics.

---

> > > ### Author Response · Authors · 2023-08-21
> > >
> > > **Motivation**
> > >
> > > To reiterate, our key motivation is to integrate diffusion and pre-trained language models into a single text generation system to combine the strengths of diffusion and AR generation. All current methods train diffusion models from scratch and we are the first to investigate the possibility of combining existing auto-regressive models with diffusion models. Our experiments in class-conditional generation show that our framework is more effective than a fine-tuned GPT-2 model. Our oracle sequence-to-sequence results also demonstrate that diffusion models have excellent coverage over the hypothesis space and have the potential to outperform auto-regressive models.
> > >
> > > **Comparison with Diffusion-LM**
> > >
> > > In our work, we focused directly on improving the language generation capabilities of diffusion language models. We evaluated our framework’s capabilities across unconditional, class-conditional, and seq2seq language generation generation. This represents a more diverse collection of datasets and settings than the Diffusion-LM paper.
> > >
> > > We compare directly against Diffusion-LM on the ROCStories dataset that they utilized for unconditional generation. We selected the ROCStories dataset primarily because it was used in their work. Our method achieves a substantially higher MAUVE score of 0.716 vs 0.043 for Diffusion-LM on that dataset (as seen in Table 2). DiffuSeq also represents a direct extension of Diffusion-LM to the Seq2Seq setting, and we compare against that method on multiple datasets (Table 6). We believe that the sample generations (provided in the appendix) highlight our improvements in coherence over Diffusion-LM and its extension to seq2seq tasks (DiffuSeq).
> > >
> > > Diffusion-LM did explore using external classifiers for plug-and-play control with the formulaic E2E restaurant review dataset. However, plug-and-play control is most relevant when the underlying model already generates high quality text. As a result, we and other recent work (e.g. DiffuSeq, GENIE, CDCD, DiffusionBERT) first focus on improving the language generation capabilities of the diffusion model. Please refer to our response to Reviewer LWig and SqZp for additional comparisons with these methods.
> > >
> > > As diffusion language models continue maturing in quality, exploring classifier-based control represents interesting future work. We leave comparisons to methods like PPLM and FUDGE to such future work exploring plug-and-play control.

---

> > > > ### Author Response · Authors · 2023-08-21
> > > >
> > > > **Sequence Lengths**
> > > >
> > > > In LD4LG, we perform diffusion in a compressed latent space (e.g. dimension 32 x 64). We can compress texts of different lengths into this latent space and then rely on the decoder’s ability to generate text sequences of different sizes when conditioned on the diffused latent code. We present some statistics for the lengths of 1000 random validation samples across various datasets we examined. We use simple whitespace tokenization to compute these statistics to be tokenizer-agnostic. The numbers are therefore low compared to the number of BPE tokens for the various models. We also compute the Pearson’s r Correlation between the input length and the length of the autoencoder reconstruction.
> > > >
> > > > | Dataset | Mean Length | StDev Length | Min Length | Max Length | Pearson's r Correlation (Input Length vs. Reconstruction Length) |
> > > > |-|-|-|-|-|-|
> > > > | ROCStories | 43.5 | 7.7 | 20 | 58 | > 0.99 |
> > > > | AG News | 30.0 | 8.3 | 8 | 56 | > 0.99 |
> > > > | WMT14-En | 18.8 | 10.9 | 1 | 77 | > 0.99 |
> > > >
> > > > We observe that all datasets have a variety of lengths and that the input and output lengths have near-perfect correlation (> 0.99). The autoencoder therefore effectively reconstructs input language of a variety of lengths from the fixed-size latent space. The near-perfect reconstruction performance reported in our submission (Table 1) also demonstrate this.
> > > >
> > > > We also present some examples of our autoencoder’s reconstructions for validation inputs of varying lengths from the WMT-14 English dataset.
> > > >
> > > > | Input | Reconstruction |
> > > > |-|-|
> > > > | That pleases me. | That pleases me. |
> > > > | For almost ten years the choir has been practising songs in this foreign,'soft' language, and now and then they bring them back to where they originally came from: the South of Africa. | For almost ten years the choir has been practising songs in this foreign,'soft' language, and now and then they bring them back to where they originally came from: the South of Africa. |
> > > > | Then she tells of her husband who was in the army. | Then she tells of her husband who was in the army. |
> > > > | Amid the usual futile arguments over who started it, scores of buildings have been reduced to rubble; more than 140 Palestinians, most of them civilians, and six Israelis have been killed; and, for the first time, missiles from Gaza have landed near Tel Aviv, Israel's metropolis, and the holy city of Jerusalem. | Amid the usual futile arguments over who started it, scores of buildings have been reduced to rubble; more than 140 Palestinians, most of them civilians, and six Israelis have been killed; and, for the first time, missiles from Gaza have landed near Tel Aviv, Israel's metropolis, and the holy city of Jerusalem. |
> > > > | Whether at a physical comfort, emotional or spiritual level. | Whether at a physical comfort, emotional or spiritual level. |
> > > > | It starts by the Antonia Fortress - Praetorium - where the judgement took place, and brings us along the streets of the Old Town to the Church of the Holy Sepulchre on Golgotha - the place of the crucifixion, Stone of Unction and the place of Jesus' burial. | It starts by the Antonia Fortress - Praetorium - where the judgement took place, and brings us along the streets of the Old Town to the Church of the Holy Sepulchre on Golgotha - the place of the crucifixion, Stone of Unction and the place of Jesus' burial. |
> > > >
> > > > We would like to emphasize that any limitations that our model has in terms of modeling datasets of different lengths will be shared by other diffusion and autoregressive models. Other language diffusion models, such as Diffusion-LM, generate instances of varying length by always generating a max-length sequence with a variable number of [PAD] tokens. To generate “It pleases me.”, for instance, Diffusion-LM would generate:
> > > >
> > > > “[BOS] It pleases me. [EOS] [PAD] [PAD] [PAD] [PAD] [PAD]... [PAD]”.
> > > >
> > > > Our hybrid approach represents an improvement over such methods.
> > > >
> > > > Autoregressive models often fail to extrapolate to longer sequence lengths than those observed during training [1].  This limitation would be shared by our framework, but it is not unique to our approach. While overcoming this limitation in autoregressive models is an active area of research [2], such advances could potentially be incorporated into our hybrid framework.
> > > >
> > > > [1] Kazemnejad, Amirhossein, et al. "The Impact of Positional Encoding on Length Generalization in Transformers." (2023).
> > > > [2] Press, Ofir, Noah A. Smith, and Mike Lewis. "Train short, test long: Attention with linear biases enables input length extrapolation." (2021).

---

### Official Review · Reviewer_5zaG · 2023-07-10

**Soundness:** 3 good
**Presentation:** 3 good
**Contribution:** 3 good
**Rating:** 6
**Confidence:** 4

**Summary:**

This paper introduces a novel approach to learning diffusion models in the latent space of an encoder-decoder language autoencoder. The proposed latent diffusion model utilizes continuous latent codes to generate text through sampling, leveraging a pre-trained decoder. The effectiveness of this approach is demonstrated across various unconditional and conditional sequence generation tasks.


**Strengths:**

1. The idea presented in the paper is straightforward and easy to understand. It extends the concept of latent diffusion models (LDM) used in other domains, such as images, to the language domain, showcasing the versatility of this approach.
2. This paper stands out by incorporating an autoregressive decoder into diffusion models, which combines the strengths of both methods. This integration significantly enhances the quality of text generation, distinguishing it from existing diffusion-based language models.


**Weaknesses:**

The basic idea of learning diffusion models in the latent space has been well explored in the image domain. Adding a self-conditioning trick has also been used in previous diffusion-LM literatures. Therefore the overall novelty is relatively limited.

**Questions:**

1. In line 107, if the latent vector is normalized by its norm, then the latents are not Gaussian anymore but become a point on the hypersphere. Will it make more sense to divide the latent with its standard deviation estimated from the data?

2. From the encoder to latents and the latents to the decoder involve two cross-attention, will this cause training instability when optimizing things end to end? On the other hand, how does the model avoid copying all encoder information to the target side?

3. How are the length and dimensions of the latents decided? Will it be possible to design variable lengths of latents to account for the actual sequence length?

**Limitations:**

The limitations were addressed.

---

> ### Author Rebuttal · Authors · 2023-08-10
>
> **Novelty**
>
> While we agree latent diffusion is commonly used for images, the adaptation to language is non-trivial and presents unique challenges given the discreteness of the domain. Latent diffusion models are reliant on the strength of the underlying autoencoder, and we introduce a method for effectively and efficiently adapting existing pre-trained language models into strong language autoencoders. The discreteness of the modality is another challenge and, as the reviewer acknowledges, we are distinctive from other diffusion language models in developing a hybrid approach utilizing the complementary strengths of diffusion and autoregressive generation.
>
> **Latent Normalization**
>
> It is not critical that the samples from the data distribution are approximately Gaussian. For instance, diffusion models are commonly trained in the pixel-space of natural images. It is, however, beneficial to control the scale of the data samples to ensure that common noise schedules can be applied without adjustment. The reviewer is correct to point out that re-scaling by the standard deviation would serve a similar purpose as our L2 normalization.
>
> In our preliminary experiments, we experimented with both enforcing L2-normalization during training and dividing the latents by an empirical estimate of the standard deviation as the reviewer suggests; we found that both were effective. A key technique employed by image diffusion models is to rescale pixel values after each diffusion step to the range of allowed values (i.e. [-1,1]). Such techniques have been shown to be critical for utilizing large guidance weights during inference with text-to-image diffusion models (e.g. Imagen). One benefit of using L2 normalization is that it enables an analogous practice of clamping the estimated data to the manifold by re-normalizing the estimate of the data after each diffusion step.
>
> We observed that the L2 normalization approach did make the downstream diffusion model much more robust to strong guidance during generation as expected. To illustrate this we report the perplexity of the generated XSum summaries across guidance weights for the two methods from some of our earlier experiments exploring this choice.
>
> |  | Autoencoder w/ STD Re-Scaling | Autoencoder w/ l2 Normalization |
> |-|-|-|
> | Guidance Scale | Perplexity | Perplexity |
> | 1.0 | 75 | 78 |
> | 2.0 | 69 | 77 |
> | 5.0 | 136 | 94 |
> | 10.0 | 351 | 123 |
>
> We observed that large guidance scales break the generated language when using STD re-scaling, but this is mitigated by using L2 normalization with the clamping trick from image diffusion. In practice, we found that relatively modest guidance (w=2.0) maximized performance for the datasets we examined (see Figure 3), so this particular design decision did not end up being critical for the performance of our approach. We will add an ablation of this decision to the appendix with discussion of our observations; it will be relevant for settings where strong guidance is necessary and would likely be of interest to the community.
>
> **Cross-attention/ Training stability**
>
> The Perceiver Resampler, which we utilize for the language autoencoder, was originally introduced to compress image features for integration into the vision-language model, Flamingo [1]. Similar to our setup, the compressed image features cross-attend to the output of a ViT encoder and are then cross-attended by the language decoder. Flamingo therefore has a similar sequence of cross-attention layers and is stable at scale (80 billion parameters).
>
> Empirically, we did not observe training stability to be a problem in our setting. We tuned the initial parameters such as the learning rate for the ROCStories dataset and were able to apply them directly to all other datasets without adjustment.
>
> During the autoencoder training, the Perceiver Resampler bottlenecks the variable-length encoder feature representation to a 2048-d latent, an over 20x reduction of the encoder features. This prevents directly copying the encoder features and forces the model to learn a compact representation of the language that can be reconstructed by the decoder. Our strong performance across diverse datasets indicates that the autoencoder is effectively generalizing beyond memorization.
>
> [1] Alayrac, Jean-Baptiste, et al. "Flamingo: a Visual Language Model for Few-Shot Learning." (2022).
>
> **Length of the latents**
>
> For the size of the latents, our goal was to maximize compression while maintaining strong reconstruction performance with the auto-encoder. This accelerates learning for the diffusion process while maintaining the capacity for high-fidelity language generation. We observed that 32x64 dimensional latents led to high quality reconstructions across all datasets and the reduction in sequence length accelerated model training. When ablating the size of the latents for the auto-encoder, we observed that further decreasing the dimensionality of the latents (e.g. 16x64 or 32x32) began to degrade reconstruction performance. We will add this ablation to the supplementary material. We would also like to re-emphasize that our method can generate sequences of different length; even though the size of the latent that is used to condition the decoder is fixed, we generate text using the decoder until the EOS token is generated.

---

> > ### Comment · Reviewer_5zaG · 2023-08-18
> >
> > Thank you for the detailed response. I will keep my score as 6.

---

### Official Review · Reviewer_SqZp · 2023-07-11

**Soundness:** 3 good
**Presentation:** 2 fair
**Contribution:** 2 fair
**Rating:** 5
**Confidence:** 4

**Summary:**

This paper introduces a latent diffusion model for natural language generation, utilizing a continuous diffusion framework. The authors leverage pretrained encoder-decoder models, such as BART and Flan-T5, to establish a continuous feature space. They propose a Perceiver Resampler as a compression module to handle variable-length and high-dimensional challenges, allowing effective learning of the diffusion process and parameterized denoising. The generative process includes a reconstruction module to map the low-dimensional latent space back to the feature space and is supported in unconditional, class-conditional, and seq2seq settings. The proposed approach demonstrates its effectiveness through extensive experiments.

**Strengths:**

1. The utilization of pretrained encoder-decoder models to establish a feature space is a clever and efficient approach, distinguishing it from previous work that relied on less-capable embedding "encoder-decoder" for latent diffusion.
2. The paper conducts extensive experiments in various settings, including unconditional, class-conditional, and seq-conditional text generation. The evaluation metrics cover aspects of quality, diversity, and other relevant factors.

**Weaknesses:**

1. Evaluation concerns arise from the lack of a multilingual setting in the benchmark datasets investigated. The benchmark datasets investigated in the paper were various, which was good, but lacked of multilingual setting, making the evaluation not fully convincing. Several previous diffusion-based text generation studies experimented on machine translation, e.g, CDCD [2], Difformer [2], and DINOISER [3] studied on WMT14 English-to-German dataset, which is most commonly-used setting for non-autoregressive text generation. Can the proposed LD4LG model also be applied in a multilingual setting?

2. The paper lacks a comprehensive literature review, hence comparison and discussion on its contributions with relevant studies. In the experiments, only diffusion-lm and diffuseq were compared. Additionally, the related work section primarily cites papers from 2022, while significant progress has been made in text diffusion research this year. What are the advantages of LD4LG compared to other continuous diffusion models, such as CDCD, Difformer, and DINOISER? How does it compare to discrete diffusion models like DiffusionBERT, reparameterized discrete diffusion model (RDM), and DiffusER?

---
[1] Continuous diffusion for categorical data

[2] Difformer: Empowering Diffusion Model on Embedding Space for Text Generation

[3] DINOISER: Diffused Conditional Sequence Learning by Manipulating Noises

[4] DiffusionBERT: Improving Generative Masked Language Models with Diffusion Models

[5] A Reparameterized Discrete Diffusion Model for Text Generation

[6] DiffusER: Discrete Diffusion via Edit-based Reconstruction

**Questions:**

See weaknesses.

I am happy to raise my rating upon the authors' responses to my concerns.

---

> ### Author Rebuttal · Authors · 2023-08-10
>
>
> **Multilingual Evaluation:**
>
> We thank the reviewer for their thoughtful feedback on our paper. We agree that evaluating our approach on a multilingual dataset would further demonstrate the versatility of our approach. The extension of our method to such settings is straightforward given the availability of strong pre-trained multilingual language models. We conducted additional experiments on the WMT14 German->English translation task following the same setup as our other Seq2Seq datasets utilizing a Multilingual T5 model. We report the reconstruction performance below for both English and German for the MT5-base language autoencoder. We observe that it can similarly be adapted to highly effective autoencoders across diverse languages.
>
> Model | Data | ROUGE-1/2/L | BLEU |
> |-|-|-|-|
> | MT5 Autoencoder | WMT14 English | 99.7/99.2/99.7 | 99.2 |
> | MT5 Autoencoder | WMT14 German | 99.8/99.4/99.8 | 99.1 |
>
> We can then train our latent diffusion model for machine translation similarly to our other Seq2Seq datasets. We present our preliminary results below and compare against the additional provided references that report results for that dataset. Unfortunately, the Difformer and DiffusER papers do not open source their implementation or report sufficient details about their evaluation setting to ensure an apples-to-apples comparison. For instance, the Difformer paper reports a tokenized BLEU score as their primary metric but does not provide their tokenizer, or sufficient details to reproduce their tokenization. It is well established that such choices significantly impact BLEU scores and must be controlled for valid comparison [1] [2]. We therefore compare against the methods that follow best practices and report reproducible SacreBLEU scores on de-tokenized generations to ensure that the comparison is valid. We also report the training durations for each method based on the reported training settings (or based on the official codebase in the case of DiNOISER).
>
> The WMT14 De-En dataset, with approximately 4.5 million training instances, is orders of magnitude larger than the other datasets we examined. Given its immense size, maximizing performance typically requires significantly more computational resources than we utilized for training the models in our work. We began training our model as quickly as we were able, but our method is still significantly undertrained compared to the related work due to the limited time available for our response. We present preliminary results below for WMT14 De->En.
>
> | Model | SacreBLEU |
> |-|-|
> | Diffusion-LM (n=5) | 20.3 |
> | CDCD (n=1) (~227.6 epochs) | 24.9 |
> | DiNOISER (n=5) (1000 epochs) | 28.8 |
> | LD4LG (MT5) (n=1) (14.4 epochs) | 24.1 |
>
> Our method again outperforms the Diffusion-LM baseline (adapted for MT in the DiNOISER paper). Even when training our model on 6.3% as much data as the CDCD model, its performance is within a single BLEU point. This demonstrates that our presented framework can be effectively extended to multilingual tasks such as machine translation. Our model does lag meaningfully behind DiNOISER with MBR decoding, but given that it is trained on less than 1.5% as much data, it is difficult to compare the approaches directly.
>
> Our model is still improving steadily with training, so the presented result is a loose lower bound on the effectiveness of our approach. We will include finalized results for WMT14 En-De for both translation directions in our camera ready.
>
> [1] A Call for Clarity in Reporting BLEU Scores
>
> [2] Scientific Credibility of Machine Translation Research: A Meta-Evaluation of 769 Papers
>
> **Literature Review:**
>
> We thank the reviewer for providing additional references. We focused our comparisons on the state-of-the art diffusion approaches published when we began the project, but acknowledge that several preprints describing concurrent language diffusion methods have since been posted. We would like to note that our DiffuSeq baseline was published at ICLR 2023, which occurred the same month as the submission deadline. We, however, acknowledge that rapid progress is being made in the area and will expand our related work discussion to include the provided references and compare directly against these methods when possible. For the methods that utilize the same paraphrasing dataset, QQP, we report the performance comparison below, and we will add these results to the camera-ready. We note that these methods use a different BERT model to compute BERTScore, so we recompute BERTScore with that model.
>
> | Method | MBR | ROUGE-L | BERTScore |
> |-|-|-|-|
> | DiffuSeq | n=10 | 58.80 | 83.65 |
> | RDM-absorbing | n=1 | 57.89 | 83.74 |
> | RDM-multinomial | n=1 | 57.25 | 83.66 |
> | RDM-absorbing | n=10 | 59.45 | 84.72 |
> | RDM-multinomial | n=10 | 58.86 | 84.66 |
> | DiffusionBERT | n=10 | 58.45 | - |
> | LD4LG (BART) | n=1 | 60.3 | 85.82 |
> | LD4LG (T5) | n=1 | 59.7 | 85.79 |
> | LD4LG (BART) | n=5 | **61.1** | **86.18** |
> | LD4LG (T5) | n=5 | 60.7 | 86.07 |
>
> We observe that LD4G outperforms all other methods using both BART and FLAN-T5, even when n=1. We believe that our approach has two key advantages over these baselines. First, by utilizing existing pre-trained language models to establish a high-quality continuous feature space, we are still able to take advantage of the strong language representations obtained from self-supervised pre-training. Second, our hybrid approach better exploits the strengths of the two complementary generative modeling paradigms. By offloading the challenge of modeling a discrete distribution to the autoencoder, we simplify the diffusion process by restricting it to the continuous, latent feature space.
>
> In light of our extended comparisons and experiments, we would like to ask if you are willing to reconsider your score. We are happy to discuss any additional concerns or questions during the discussion period!

---

> > ### Comment · Reviewer_SqZp · 2023-08-15
> > **Thanks for your response and increase my rating**
> >
> > Dear authors,
> >
> > Thank you so much for your efforts in providing thorough results and discussions! Although there're still rooms for improvement for multilingual/machine translation given the WMT14 De-En results, the responses have mostly addressed my concerns in general. I accordingly raise my rating from 4 to 5. Please add the new results and discussions in the revision.

---

> > > ### Author Response · Authors · 2023-08-21
> > > **Updated Machine Translation Results**
> > >
> > > Thank you for updating your score! We provide updated machine translation results below with further training.
> > >
> > > | Model | SacreBLEU |
> > > |-|-|
> > > | Diffusion-LM (n=5) | 20.3 |
> > > | CDCD (n=1) (~227.6 epochs) | 24.9 |
> > > | CDCD (n=10) (~227.6 epochs) | 25.4 |
> > > | DiNOISER (n=5) (1000 epochs) | 28.8 |
> > > | LD4LG (MT5) (n=1) (~28 epochs) | 26.5 |
> > > | LD4LG (MT5) (n=5) (~28 epochs) | 27.0 |
> > >
> > > We observe that our method comfortably outperforms CDCD while training on 15% as much data. The updated result also more than halves the distance between our prior result and DiNOISER while training on less than 3% as much data. The strong performance of LD4LG with comparably little training demonstrates the value of incorporating strong pre-trained models. Our method is still steadily improving with training and is therefore still undertrained compared to the baselines.
> > >
> > > We look forward to incorporating your thoughtful feedback into the final version of the paper. Thank you again for engaging with our response!

---

### Author Rebuttal · Authors · 2023-08-10


We would like to thank the reviewers for their thoughtful feedback on our paper. In particular, we appreciate Reviewer SqZp highlighting that the “utilization of pretrained encoder-decoder models to establish a feature space is a clever and efficient approach”. We also thank Reviewer 5zaG for noting our submission “stands out by incorporating an autoregressive decoder into diffusion models, which combines the strengths of both methods” and “significantly enhances the quality of text generation, distinguishing it from existing diffusion-based language models.” As reviewers T1gz and TpnG point out, we perform comprehensive experiments across unconditional, class-conditional and sequence-to-sequence settings to demonstrate the effectiveness of our method.

We have responded to each reviewers' valuable critiques in our individual responses and look forward to incorporating their feedback into the final version. We hope to continue this valuable discussion during the discussion period.

We also present a general comment here regarding the motivation, long-term potential, and contribution of this work to address points brought up by Reviewers T1gz and LWig.

We believe diffusion models provide complementary strengths to autoregressive generation in areas like sample diversity and controllability. Our latent diffusion approach was more effective at generating novel text and more effective at topic-guided generation than GPT-2, an autoregressive model with nearly double the trainable parameters as our diffusion model. For the Seq2Seq tasks, our oracle evaluation demonstrated that drawing just 5 samples from the diffusion language model consistently produced a higher-quality generation than that produced by the fine-tuned autoregressive language model. This demonstrates that the diffusion model has excellent coverage over the hypothesis space and that further exploration of re-ranking techniques could lead them to consistently outperform the fine-tuned language models.

Although latent diffusion is commonly used for images, its extension to language presents unique challenges given the discreteness of the domain. Latent diffusion models are reliant on the strength of the underlying autoencoder, and we introduce a method for effectively and efficiently adapting existing pre-trained language models into strong language autoencoders. As multiple reviewers acknowledge, we are distinctive from other diffusion language models in developing a hybrid approach utilizing the complementary strengths of diffusion and autoregressive generation.

As for the potential for impact, we believe that language diffusion for language is still in its early days as a paradigm, with research needed to realize its full potential. As a point of comparison, diffusion models did not surpass GANs on image synthesis benchmarks until 6 years after their original introduction. They have since become the dominant paradigm in that domain, though whether language diffusion models will follow a similar trajectory remains to be seen. We hope our work provides a foundation for further research into language diffusion as a promising new paradigm. In this work, we are the first to demonstrate that diffusion models can be integrated with existing pre-trained language models for high quality language generation.

---

### Decision · Program_Chairs · 2023-09-21

**Decision:**

Accept (poster)

**Comment:**

This paper presents an interesting approach to combine diffusion models with encoder-decoder language models in order to learn high-quality language autoencoders. The authors demonstrate the effectiveness of continuous diffusion models in the latent space of the language autoencoder, resulting in continuous latent representations that can be decoded into natural language. The proposed method is validated across various language generation tasks and diverse datasets, showing superior performance compared to previous diffusion language models. For reviewer T1gz's concern, I totally understand it but think the motivation question is a common challenge for the research on diffusion model topics. This does not make it less interesting for a wider group of audience in the community. Therefore, I would recommend to accept this paper.